# Extrusion-modulated DnaA activity oscillations coordinate DNA replication with biomass growth

**Dengjin Li[1,2†], Hai Zheng[1,2†], Yang Bai[1,2†], Zheng Zhang[1], Hao Cheng[1], Xiongliang Huang[1,2], Ting Wei[1], Matthew Chang[3], Arieh Zaritsky[4], Terence Hwa[5], Chenli Liu[1,2]***

[1]Key Laboratory of Quantitative Synthetic Biology, Shenzhen Institute of Synthetic Biology, Shenzhen Institutes of Advanced Technology, Chinese Academy of Sciences, Beijing, China; [2]University of Chinese Academy of Sciences, Beijing, China; [3]NUS Synthetic Biology for Clinical and Technological Innovation and Department of Biochemistry, Yong Loo Lin School of Medicine, National University of Singapore; National Centre for Engineering Biology, Singapore, Singapore; [4]Faculty of Natural Sciences, Ben-Gurion University of the Negev, Beersheba, Israel; [5]Department of Physics & Department of Molecular Biology, University of California at San Diego, San Diego, United States

**\*For correspondence:**
cl.liu@siat.ac.cn

[†]These authors contributed equally to this work

**Competing interest:** The authors declare that no competing interests exist.

## eLife Assessment

This work provides high-precision single-cell data on the relationship between DnaA activity and cell size, offering **important** insights for the field of cell cycle control. These findings motivate a novel and intriguing hypothesis for DNA replication initiation -the "extrusion model"- in which DNA-binding proteins modulate free DnaA availability in response to biomass-DNA imbalance. While the current indirect evidence does not fully establish the model, an experimental perturbation involving H-NS offers **convincing** support for its plausibility, laying the groundwork for future investigation.

**Abstract** Robust control of DNA replication is fundamental to bacterial proliferation. In *Escherichia coli*, replication initiation is thought to be regulated by oscillations in DnaA activity, driven by DnaA-chromosome interactions that differ among leading models. However, direct evidence linking these oscillations to replication initiation has been lacking, and existing models fail to explain the observed decoupling of replication initiation from *dnaA* expression. Here, we establish a direct link between DnaA activity and replication initiation by demonstrating robust oscillations in DnaA activity, which peak precisely at replication initiation across diverse growth conditions and genetic perturbations. Notably, these oscillations persist even when *dnaA* transcription remains constant, suggesting a regulatory mechanism that modulates DnaA activity independently of its expression. Additionally, we propose an extrusion model in which DNA-binding proteins sense biomass-DNA imbalance and extrude DnaA from the chromosome to trigger replication, overcoming limitations of existing models. Consistent with this model, perturbation of the nucleoid-associated protein H-NS modulates DnaA activity and replication timing, supporting its mechanistic validity.

## Introduction

Bacterial proliferation requires precise coordination between DNA replication and biomass growth to maintain cellular homeostasis. While biomass accumulates exponentially, DNA synthesis proceeds in a stepwise manner, creating temporal imbalances that demand continuous correction throughout the cell cycle (*Figure 1A and B*). Failure to synchronize these processes risks catastrophic outcomes, such as growth defects, incomplete replication, unequal genetic segregation, and cell death (*Boye et al., 1996*; *Khodursky et al., 2015*; *Lu et al., 1994*; *Mäkelä et al., 2024*; *Zaritsky and Pritchard, 1971*). Central to this synchronization in *Escherichia coli* is the initiator protein DnaA, a DNA-binding protein that binds specifically to an asymmetric 9 bp DnaA-box, whose consensus sequence is TTWTNCACA (*Hansen and Atlung, 2018*; *Speck et al., 1999*). Although DnaA binding to the replication origin (*oriC*) triggers replication initiation (*Dong et al., 2023*; *Kaguni, 2011*; *Ohbayashi et al., 2020*; *Ozaki and Katayama, 2012*; *Sekimizu et al., 1987*), how DnaA activity—defined as the capacity to initiate replication—is dynamically regulated to coordinate DNA synthesis with cellular growth, particularly when DNA content and biomass diverge, remains a fundamental unresolved question in cell cycle biology.

Two competing models have dominated the field. The titration model posits that DnaA activity is governed by the availability of free DnaA, which accumulates until it surpasses the buffering capacity of chromosomal DnaA-boxes (*Christensen et al., 1999*; *Hansen et al., 1991*). Here, replication initiation resets the system by increasing DNA content (and thus DnaA-boxes), titrating excess DnaA. In contrast, the switch model emphasizes the ATP/ADP-bound states of DnaA: DnaA-ATP, synthesized during biomass growth, triggers initiation, while hydrolysis to DnaA-ADP following replication resets the cycle (*Berger and Wolde, 2022*; *Donachie and Blakely, 2003*; *Fu et al., 2023*; *Hansen and Atlung, 2018*; *Katayama et al., 2017*). Crucially, the titration model depends on ongoing DnaA synthesis to reflect biomass accumulation, whereas the switch model relies on nucleotide-state transitions, independent of total DnaA levels. Despite these mechanistic differences, both models consider that replication initiation is regulated by oscillations in DnaA activity, driven by DnaA-chromosome interactions (*Figure 1C*). However, direct evidence linking these oscillations to replication initiation remains lacking. Moreover, neither model fully explains recent experimental observations: replication initiation persists for multiple generations even when *dnaA* transcription is completely inhibited (*Knöppel et al., 2023*). This discrepancy suggests an uncharacterized mechanism that senses biomass-DNA imbalance and directly modulates DnaA activity.

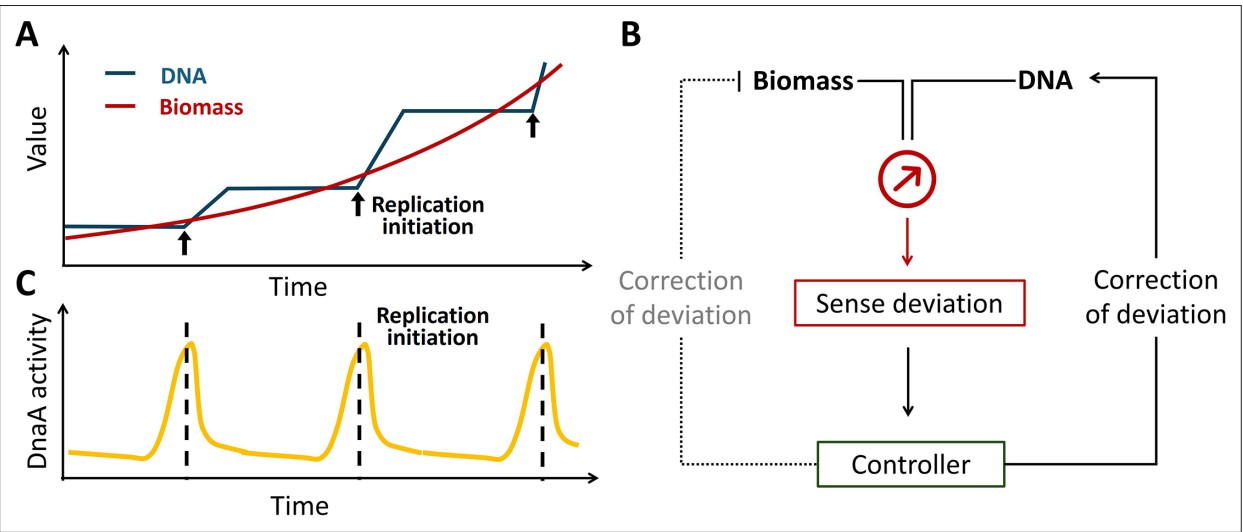

**Figure 1.** Coordination of biomass growth and DNA replication through replication initiation and DnaA activity oscillations. (**A**) Schematic representation of the discrepancy between biomass accumulation and DNA replication. While biomass grows exponentially, DNA synthesis progresses linearly, necessitating replication initiation events to maintain coordination. (**B**) Mechanistic model for biomass-DNA coordination in bacteria. A molecular sensor detects deviations between cell mass and DNA content, transmitting this information to regulatory controllers that compensate by either increasing DNA replication or restricting biomass accumulation. (**C**) Illustration of cyclic DnaA activity oscillations aligning with replication initiation to ensure precise cell cycle control.

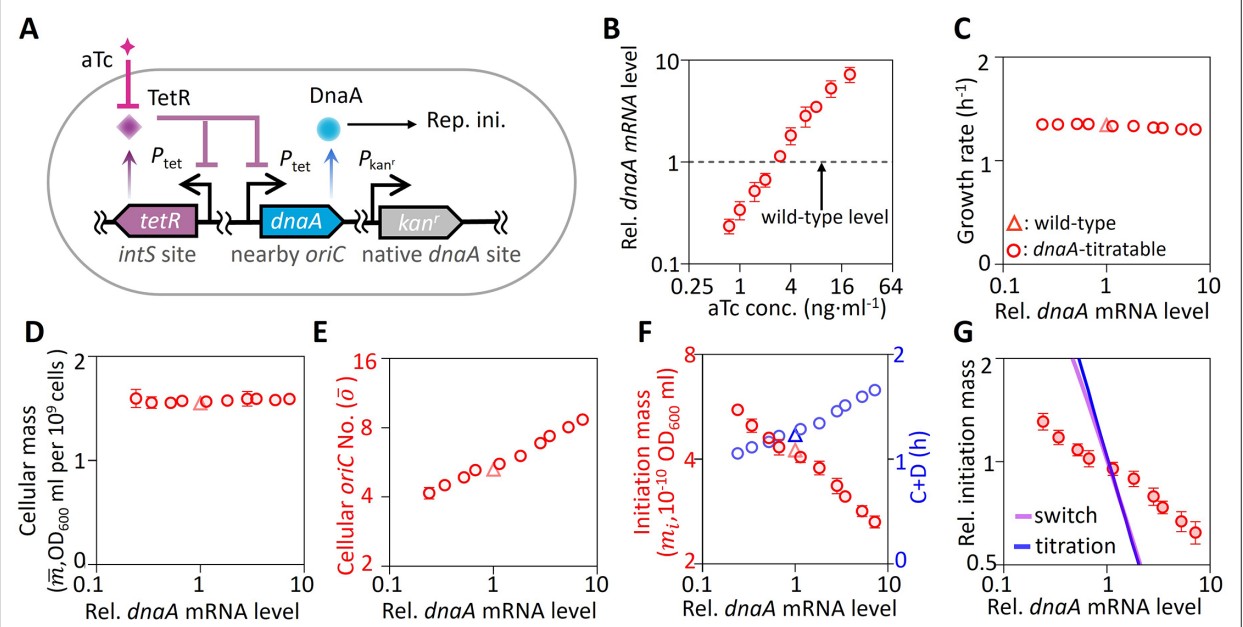

**Figure 2.** Construction and characterization of the *dnaA*-titratable strain. (**A**) Schematic of the *dnaA*-titratable strain. A *dnaA* gene under the control of the $P_{tet}$ promoter was inserted near *oriC* and regulated by a $P_{tet}$-*tetR* feedback loop integrated at the *intS* locus, enabling fine-tuned expression control. The native *dnaA* gene was replaced with a kanamycin resistance cassette (*kan'*). (**B–F**) Characterization of *dnaA*-titratable cells (circle) and wild-type MG1655 cells (triangle) grown in rich defined medium with glycerol (M6) under varying aTc concentrations. Measured parameters include: (**B**) *dnaA* mRNA levels; (**C**) growth rate; (**D**) population-averaged cellular mass; (**E**) population-averaged *oriC* numbers; (**F**) initiation mass (red, left axis); and the initiation to division period (C+D) (blue, right axis). The *dnaA* mRNA levels were normalized to that in wild-type cells. Cellular mass was determined by $OD_{600}$ divided by cell number concentration. *oriC* copy numbers were measured using a run-out assay, and initiation mass was calculated as the ratio of cellular mass to *oriC* numbers. Data represent means ± SD ($n = 5$ biological replicates). (**G**) Relationship between relative initiation mass and relative *dnaA* mRNA levels, compared with predictions from the initiation titration model (blue line) and the switch model (purple line). The relative *dnaA* mRNA levels in experiments are compared to relative DnaA expression rate $\alpha_A$ in models. Experimental data are overlaid for validation.

The online version of this article includes the following source data for figure 2:

**Source data 1.** Source data for *Figure 2* showing the physiological characteristics and model predictions of the *dnaA*-titratable cells under different *dnaA* expression levels.

Here, using a synthetic reporter coupled with single-cell mRNA fluorescence in situ hybridization (FISH), we demonstrate that DnaA activity—governed by free DnaA concentration, DnaA-ATP/-ADP ratio, and orisome assembly competence—peaks precisely at initiation across diverse growth conditions and genetic backgrounds, even when *dnaA* transcription is held constant. DnaA activity oscillations persist independently of transcriptional feedback, pointing to a posttranslational regulatory mechanism. We propose an extrusion model in which nucleoid-associated proteins (NAPs) sense biomass-DNA imbalance and dynamically extrude DnaA from the chromosome, liberating it to activate *oriC*. This model not only explains the decoupling of replication timing from *dnaA* expression but also integrates disparate experimental findings. Perturbations of the NAP, H-NS, for instance, modulate DnaA activity and replication timing, directly supporting the model's predictions. By unifying regulatory principles across transcription, protein activity, and chromosome dynamics, our findings advance the paradigm for bacterial replication control.

## Results

### Development of a DnaA activity reporter system

It has been known that changing DnaA expression levels affects DnaA activity and further changes initiation mass (**Berger and Wolde, 2022**; **Løbner-Olesen et al., 1989**; **Zheng et al., 2020**). To investigate this further, we engineered a *dnaA*-titratable strain, where *dnaA* transcription is controlled by an inducible $P_{tet}$ promoter (**Lutz and Bujard, 1997**; **Figure 2A**). This system enabled precise modulation of *dnaA* mRNA levels (0.25–6 times the wild-type level) via anhydrotetracycline (aTc) induction

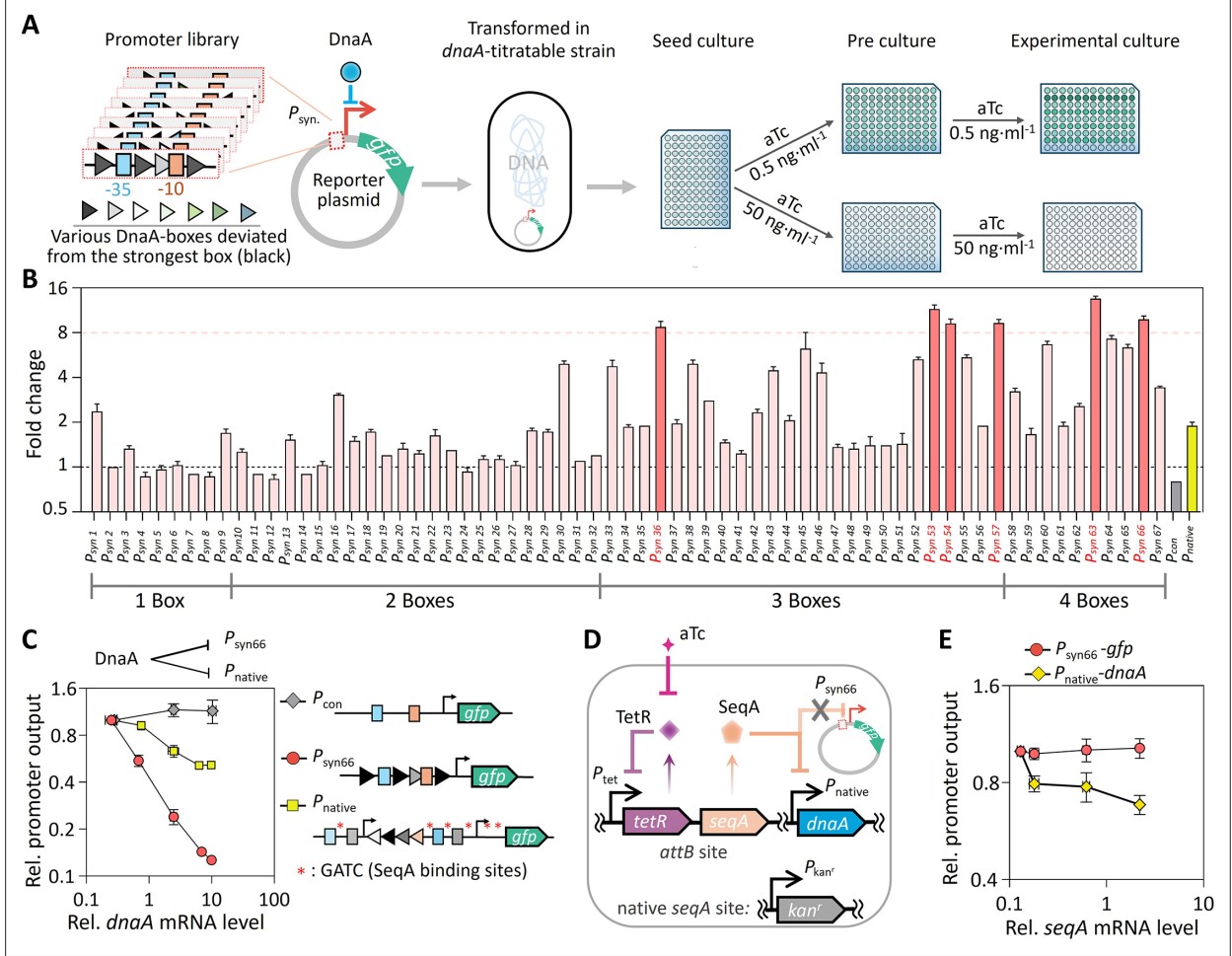

**Figure 3.** Development of a DnaA activity reporter system. (**A**) Schematic of promoter design and screening. Sixty-seven synthetic promoters were constructed by inserting various DnaA-boxes around the promoter core to drive *gfp* expression in a *dnaA*-titratable strain, where *dnaA* expression was regulated by aTc concentration. After pre-cultivation, GFP fluorescence per OD$_{600}$ was measured using a microplate reader in cells grown under low (0.5 ng·ml$^{-1}$) or high (50 ng·ml$^{-1}$) aTc concentrations to determine repression fold-change. (**B**) Repression fold-change of synthetic promoters. $P_{con}$ (a promoter lacking DnaA-boxes) and $P_{native}$ (the endogenous *dnaA* promoter) served as negative and positive controls, shown in gray and yellow, respectively. (**C**) Response curves of three promoters to varying *dnaA* expression levels, with their promoter architectures shown on the right. Promoter activity was assessed by relative *gfp* mRNA levels, normalized to the lowest *dnaA* expression condition. Schematic (**D**) and response curves (**E**) of $P_{syn66}$ and $P_{native}$ responses to SeqA in the *seqA*-titratable strain. Promoter activity was quantified from *gfp* transcript levels in *seqA*-titratable cells containing the $P_{syn66}$-GFP plasmid, normalized to the lowest *seqA* expression level. All cells were grown in rich defined medium supplemented with glycerol across different aTc concentrations. Data represent mean ± SD from 3 biological replicates (**B, C, E**).

The online version of this article includes the following source data and figure supplement(s) for figure 3:

**Source data 1.** Source data for *Figure 3* showing the response characteristics of the synthetic promoter under different expression levels of DnaA and SeqA proteins.

**Figure supplement 1.** Screening a library of synthetic promoters reveals potential candidates to report DnaA activity.

(0.75–20 ng·ml$^{-1}$) without altering growth rate or cell size (*Figure 2B–D*). As the DnaA expression level increases, DnaA activity reaches the initiation threshold earlier. Given that cell mass remained nearly unchanged, this earlier initiation led to an increase in population-averaged cellular *oriC* numbers (*Figure 2E*). Concurrently, the initiation mass was reduced by 50%, and the period from initiation to division (C+D) was increased by ~60% (*Figure 2F*). Notably, the observed relationship between DnaA expression and initiation mass deviated significantly from both the titration and switch models (*Figure 2G*; Appendix 1—note 1), suggesting unaccounted regulatory mechanisms.

To dissect DnaA activity dynamics, we constructed a library of synthetic promoters (*n*=67) by replacing *tetO* operators in $P_{tet}$ with DnaA-boxes (*Figure 3—figure supplement 1*). To assess the

degree of DnaA activity-mediated repression, we measured the fold change in GFP expression driven by each synthetic promoter under low and high levels of *dnaA* induction (*Figure 3A*). Six synthetic promoters exhibited over eightfold DnaA-mediated repression, whereas the promoter lacking DnaA-boxes ($P_{con}$) showed none, and the well-characterized native *dnaA* promoter ($P_{native}$) (*Braun et al., 1985*; *Saggioro et al., 2013*; *Speck et al., 1999*) displayed approximately twofold repression (*Figure 3B*). $P_{syn66}$, which contains three strong and one weak DnaA-box, exhibited >8-fold repression under high DnaA induction (*Figure 3B*). $P_{syn66}$'s design enables responsiveness to both free DnaA levels (via strong boxes) and the DnaA-ATP/DnaA-ADP ratio (via differential binding to the weak box) (*Grimwade and Leonard, 2021*; *Katayama et al., 2017*; *Speck et al., 1999*). Dose-response assays confirmed that $P_{syn66}$ was progressively repressed across a 40-fold range of *dnaA* expression (*Figure 3C*), with no interference from SeqA-mediated sequestration (*Figure 3D and E*). These results validated $P_{syn66}$ as a sensitive and specific reporter for DnaA activity.

## Decoupling DnaA activity oscillations from *dnaA* transcription fluctuations

To investigate DnaA activity dynamics throughout the cell cycle, we linked $P_{syn66}$ activity to cell size, a surrogate for cellcycle progression, using single-cell mRNA FISH (*Skinner et al., 2013*) to quantify *lacZ* mRNA levels in MG1655Δ*lac* cells harboring the $P_{syn66}$-*lacZ* plasmid (*Figure 4A*). A control strain harboring a promoter-less *lacZ* construct ($P_{neg}$) showed undetectable fluorescence signals (*Figure 4A*), confirming the specificity of *lacZ* mRNA detection. Quantitative analysis revealed that $P_{syn66}$-driven mRNA levels fluctuated ~3-fold over the cell cycle, whereas the DnaA-unresponsive constitutive promoter ($P_{con}$) exhibited stable expression (*Figure 4B*). However, fluctuations in mRNA concentration may not directly correspond to changes in DnaA activity due to global factors such as plasmid copy number and RNA polymerase concentration (*Balakrishnan et al., 2022*; *Bintu et al., 2005a*; *Bintu et al., 2005b*). To account for these variables, we used a $P_{con}$-*lacZ* plasmid to estimate global effects on mRNA concentration and calculated DnaA activity by assuming that it is inversely proportional to $k_{syn66} = \frac{[mZ]\,(\mathrm{P_{syn66}})}{[mZ]\,(\mathrm{P_{con}})}$. In wild-type cells (MG1655Δ*lac*) harboring the synthetic reporter system (*Figure 4C*), fluctuations in DnaA activity, as denoted by $k_{syn66}^{-1}$, displayed approximately a 3-fold variation across the cell cycle (*Figure 4D*), underscoring the substantial cell cycle-dependent oscillations in DnaA activity.

We then examined whether the observed fluctuations in DnaA activity were linked to oscillations in *dnaA* transcription. Comparing the dynamics of *dnaA* mRNA and DnaA activity, we found that the peak in DnaA activity coincided with the trough in *dnaA* mRNA levels (dashed line in *Figure 4D and E*). To test whether *dnaA* transcription drives these oscillations, we eliminated transcriptional fluctuations using the *dnaA*-titratable strain. Despite constant *dnaA* mRNA levels (2 ng·ml⁻¹ aTc induction), DnaA activity was retained with consistent oscillations (*Figure 4F–H*), demonstrating the decoupling of DnaA activity oscillations from transcriptional fluctuations. Our data suggest that posttranslational regulation, not transcriptional feedback, governs the DnaA activity.

## DnaA activity oscillations are tightly coupled to replication timing

To explore the relationship between DnaA activity oscillations and DNA replication initiation, we examined cells grown under various nutrient conditions supporting different doubling times (30–66 min). Significant fluctuations in DnaA activity were observed across all conditions (*Figure 5A*). We also estimated the cell volume at replication initiation ($V_i$) based on population-averaged cell volume and *oriC* number (*Bremer et al., 1979*; *Si et al., 2017*; *Zheng et al., 2016*; *Figure 5—figure supplement 1A–C*), marking this on the DnaA activity profiles. In a representative birth-to-division cell cycle (*Pountain et al., 2024*; *Figure 5—figure supplement 1D*), DnaA activity peaks consistently coincided with $V_i$, indicating a close correlation between DnaA activity and replication initiation (*Figure 5A*).

Further genetic perturbations, such as titrating DnaA expression, shifted DnaA activity fluctuations, yet the relationship between $V^*$ (the cell volume at maximal DnaA activity in a representative birth-to-division cell cycle) and $V_i$ remained locked (*Figure 5B*). To quantify this relationship, we compared $V_i$ with $V^*$, revealing a strictly proportional relationship through the origin (slope = 1.0, R²=0.98), indicating equivalence between $V^*$ and $V_i$ (*Figure 5C*). These results confirm that DnaA activity oscillations are tightly correlated with replication initiation, independent of transcriptional or nutrient perturbations.

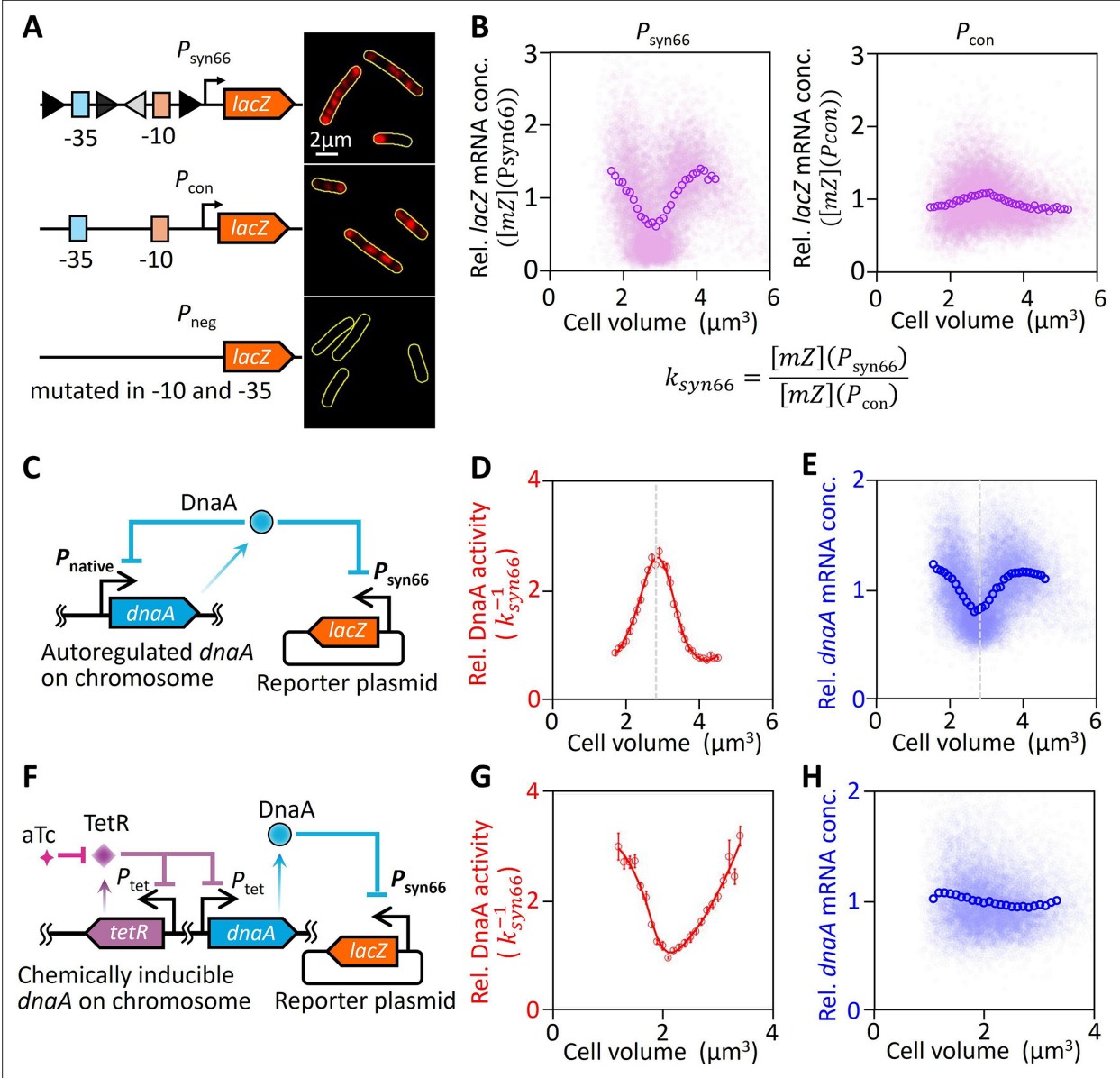

**Figure 4.** DnaA activity oscillations decoupled from *dnaA* transcription fluctuations. (**A**) Representative *lacZ* mRNA fluorescence in situ hybridization (FISH) images of MG1655 Δ*lac* cells transformed with *lacZ* expression plasmids driven by $P_{syn66}$ (DnaA-boxes around promoter core), $P_{con}$ (no DnaA-box around promoter core), or $P_{neg}$ (mutated promoter core). Yellow outlines indicate cell boundaries identified from phase-contrast images. (**B**) Relative *lacZ* mRNA concentrations driven by $P_{syn66}$ (left) and $P_{con}$ (right) across different cell volumes. Relative concentrations were determined from volume-specific *lacZ* mRNA fluorescence intensities, normalized to the population average. Volume-binned data for $P_{syn66}$ ($[mZ](P_{syn66})$) and $P_{con}$ ($[mZ](P_{con})$) are shown as open circles and were used to calculate $k_{syn66}$. (**C**) Schematic of a strain with autoregulated *dnaA* transcription carrying a DnaA activity reporter plasmid. Cell cycle-dependent fluctuations in relative DnaA activity (**D**) and relative *dnaA* mRNA concentrations (**E**) in cells from panel **C**, grown in rich defined medium supplemented with glucose. Relative DnaA activity ($k_{syn66}^{-1}$), calculated from volume-binned data in panel **B**, was smoothed and plotted as a red curve (**D**). Relative *dnaA* mRNA concentrations were determined from volume-specific *dnaA* mRNA fluorescence intensities, normalized to the population average, with volume-binned data shown as open circles (**E**). Dashed lines indicate the cell volume at peak DnaA activity (**D**) and the minimum *dnaA* mRNA content (**E**). (**F–H**) Same as panels **C–E**, but for cells with aTc induced *dnaA* transcription, grown in rich defined medium supplemented with glycerol under 2 ng·ml⁻¹ aTc induction. More than 8000 cells were analyzed per growth condition, with at least 150 cells per bin; all error bars correspond to standard error of the mean (SEM).

The online version of this article includes the following source data for figure 4:

**Source data 1.** Source data for *Figure 4* showing the changes in *lacZ* mRNA concentration driven by the reporter promoter with cell size, and the cell cycle-dependent variations in DnaA activity and *dnaA* mRNA concentration.

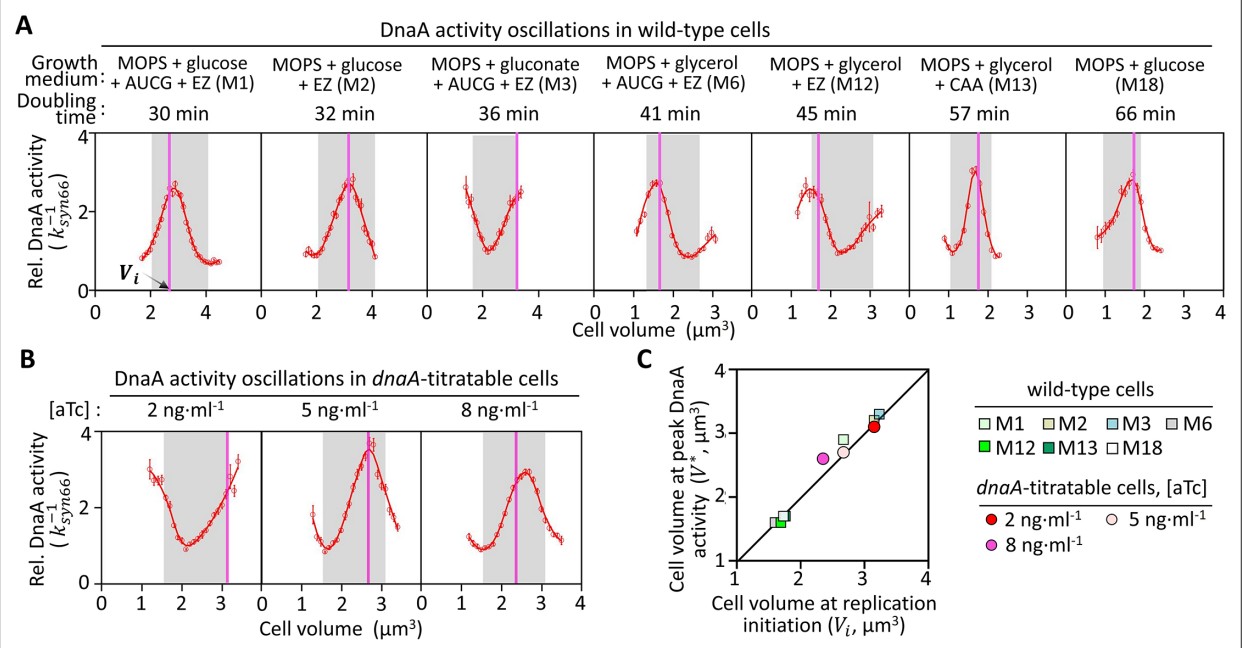

**Figure 5.** Tight correlation between DnaA activity oscillations and DNA replication initiation. (**A**) Cell cycle-dependent DnaA activity oscillations in wild-type cells across various growth conditions. DnaA activity is represented by $k_{syn66}^{-1}$. Volume-binned DnaA activity (red circles) was smoothed and plotted as a red curve. The representative birth-to-division cell cycle, defined as the cell volume doubling interval containing the majority of cells, is shaded in gray. The vertical line indicates the cell volume at replication initiation ($V_i$). (**B**) Cell cycle-dependent DnaA activity oscillations in *dnaA*-titratable cells grown in M6 medium with varying aTc concentrations. More than 8000 cells were analyzed per growth condition, with at least 150 cells per bin; error bars show the mean ± SEM. (**C**) Correlation between $V^*$ and $V_i$ in wild-type (squares) and *dnaA*-titratable (circles) cells. $V^*$ represents the cell volume at the peak of DnaA activity within the representative birth-to-division cell cycle. The black line indicates equivalence between $V^*$ and $V_i$.

The online version of this article includes the following source data and figure supplement(s) for figure 5:

**Source data 1.** Source data for **Figure 5** showing DnaA activity oscillations and DNA replication initiation in wild-type cells cultivated under various growth media and in *dnaA*-titratable cells cultivated under various induction levels.

**Figure supplement 1.** Determination of cell volume at replication initiation and representative birth-to-division cell cycle.

## An extrusion model reconciles paradoxical initiation events

Very recently, Knöppel et al. reported that multiple rounds of initiation occurred even after DnaA synthesis is shut down (*Knöppel et al., 2023*). In their experiment, DnaA was supplied in excess before expression ceased (*Knöppel et al., 2023*), potentially allowing continued replication initiation due to residual DnaA. To address this issue, we shut down DnaA expression from its native level using the deactivated miniature Un1Cas12f1 (dUn1Cas12f1) system, guided by sgRNA to repress *dnaA* transcription (*Figure 6A*). This system effectively diminished *dnaA* transcription to an extremely low level, yet total *oriC* numbers increased 4-fold within 90 min, consistent with two rounds of replication initiation (*Figure 6B*). These findings demonstrate that halting DnaA synthesis does not immediately abolish replication initiation.

Both the titration and switch models predict a close relationship between DnaA activity oscillations and replication initiation (*Berger and Wolde, 2022*). However, they fail to account for the multiple rounds of initiation observed after DnaA synthesis was shut down. According to the titration model, initiation should cease immediately once DnaA expression is shut down, as the reduction in DnaA concentration would prevent it from reaching the threshold required to trigger new rounds of replication. In contrast, the switch model predicts infinite initiations following the cessation of DnaA expression (*Figure 6C*). This is because the cessation of DnaA synthesis only reduces the production rate of DnaA-ATP and has no effect on the conversion rates between DnaA-ATP and DnaA-ADP. As a result, the ratio of DnaA-ATP to DnaA-ADP would temporarily decrease, causing a brief lag before initiation resumes indefinitely (*Figure 6C*).

The failure of both models prompts us to revisit the regulatory mechanisms of replication initiation. In the titration model, a substantial fraction of DnaA proteins binds to DnaA-boxes along the

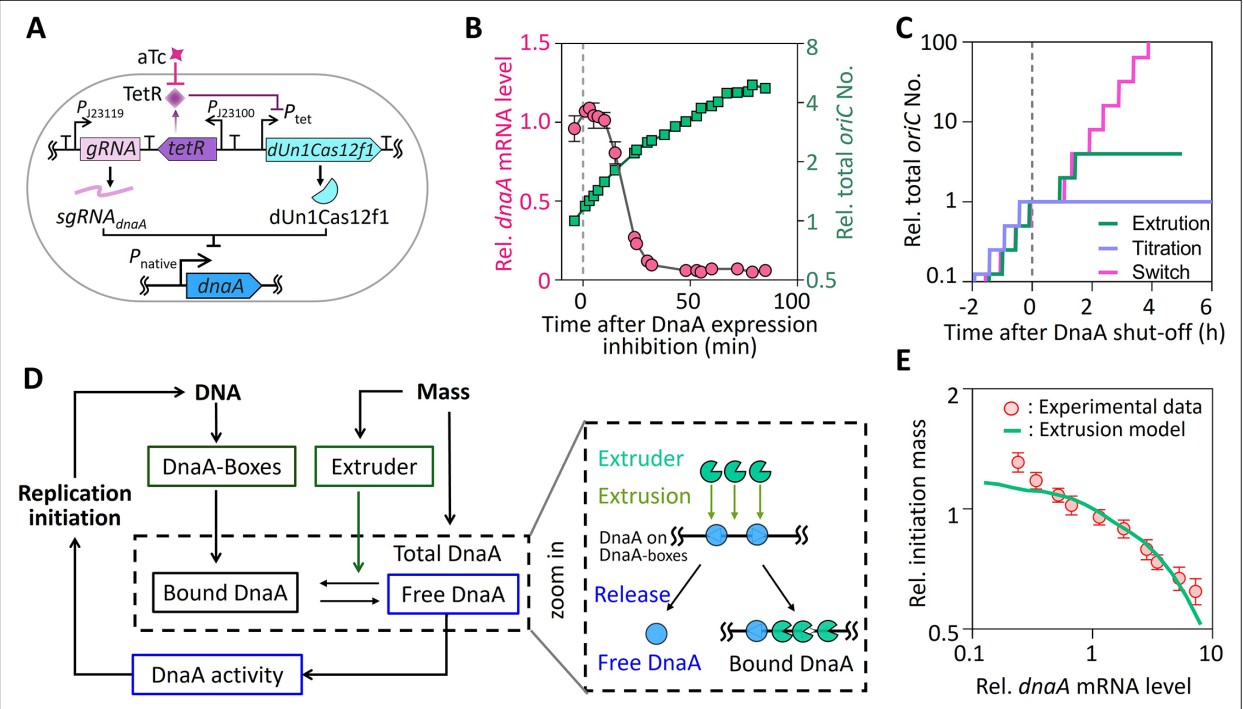

**Figure 6.** An extrusion model explains DnaA shutdown dynamics. (**A**) Genetic circuit of the deactivated CRISPR-Cas system for *dnaA* transcription shutdown. *dnaA* gene is targeted by a constitutively expressed sgRNA, while dUn1Cas12f1 expression is inhibited by TetR repressor. These transcription units are separated by terminators. The cassette was integrated into the chromosome near the *oriC* locus. DnaA shutdown is induced by the addition of aTc. (**B**) Time course of relative *dnaA* mRNA levels (red line, left axis) and total *oriC* number (green line, right axis) following the addition of 50 ng·ml⁻¹ aTc at time 0 (dashed line). *dnaA* mRNA levels were normalized to wild-type levels, and *oriC* numbers were normalized to their initial values. Error bars indicate mean ± SD (n = 3 biologically independent experiments). (**C**) Predicted increases in total *oriC* number during *dnaA* transcription shutdown based on three models: the titration model, switch model, and extrusion model. Shutdown was simulated by setting *dnaA* transcription to zero at time 0 (dashed line). (**D**) Schematic of the extrusion model. The model introduces extruder(s) as additional regulators of biomass-DNA coordination, complementing the role of DnaA (left). Increased binding of the extruder to DNA promotes the release of DnaA from DnaA-boxes (right). (**E**) Comparison of the relationship between relative initiation mass and relative *dnaA* mRNA levels from experimental data (*Figure 2F*) and predictions of the extrusion model.

The online version of this article includes the following source data and figure supplement(s) for figure 6:

**Source data 1.** Source data for *Figure 6* showing changes in DNA replication initiation after *dnaA* shutdown, as well as the extrusion model prediction regarding the relationship between initiation mass and DnaA expression level.

**Figure supplement 1.** Predictions of the extrusion model.

chromosome, raising the possibility that these bound proteins serve as a reservoir of free DnaA, sustaining replication initiation even after DnaA synthesis ceases. We thus considered the possibility of some extruder(s) alongside DnaA to sense biomass-DNA deviation and flush DnaA proteins away from the DnaA-boxes, thus releasing free DnaA to dynamically restore DnaA activity, even in the absence of ongoing DnaA synthesis (*Figure 6D*).

In this proposed 'extrusion model', the extruder is constitutively expressed, synthesized at a constant rate throughout the cell cycle, accumulating in proportion to biomass growth, enabling replication initiation when free DnaA concentrations exceed the critical threshold, as described in the titration model (*Hansen and Atlung, 2018*; *Hansen et al., 1991*; *Figure 6D*). This model guarantees the long-term coordination of biomass growth and *oriC* abundance, as well as stable oscillations in free DnaA and *oriC* concentrations (*Figure 6—figure supplement 1A*). Notably, the extrusion model accounts for the persistence of replication initiation and free DnaA oscillations over multiple cell cycles after DnaA synthesis is halted (*Figure 6C*; *Figure 6—figure supplement 1B*). Moreover, since the extruder mitigates the impact of perturbations on DnaA synthesis, the model predicts a quantitative relationship between initiation mass and DnaA expression levels that aligns with experimental observations (*Figure 6E*; Appendix 1—note 1). Furthermore, a stochastic implementation of the extrusion

model shows the lack of correlation between successive replication initiation events (*Figure 6—figure supplement 1C*), as observed in recent studies (*Si et al., 2019*; *Witz et al., 2019*).

## Perturbation of H-NS validates the extrusion mechanism

The extrusion model could be validated by changing the expression level of the extruder, which modulates DnaA activity by releasing DnaA from DnaA-boxes. Given their DNA binding properties, NAPs (*Dillon and Dorman, 2010*) are likely candidates for this extruder role. Therefore, we selected H-NS (*Dorman, 2004*), a major NAP, which is capable of promoting the release of DnaA from bound DnaA-boxes in vitro (*Figure 7—figure supplement 1*), to experimentally assess the validity of the

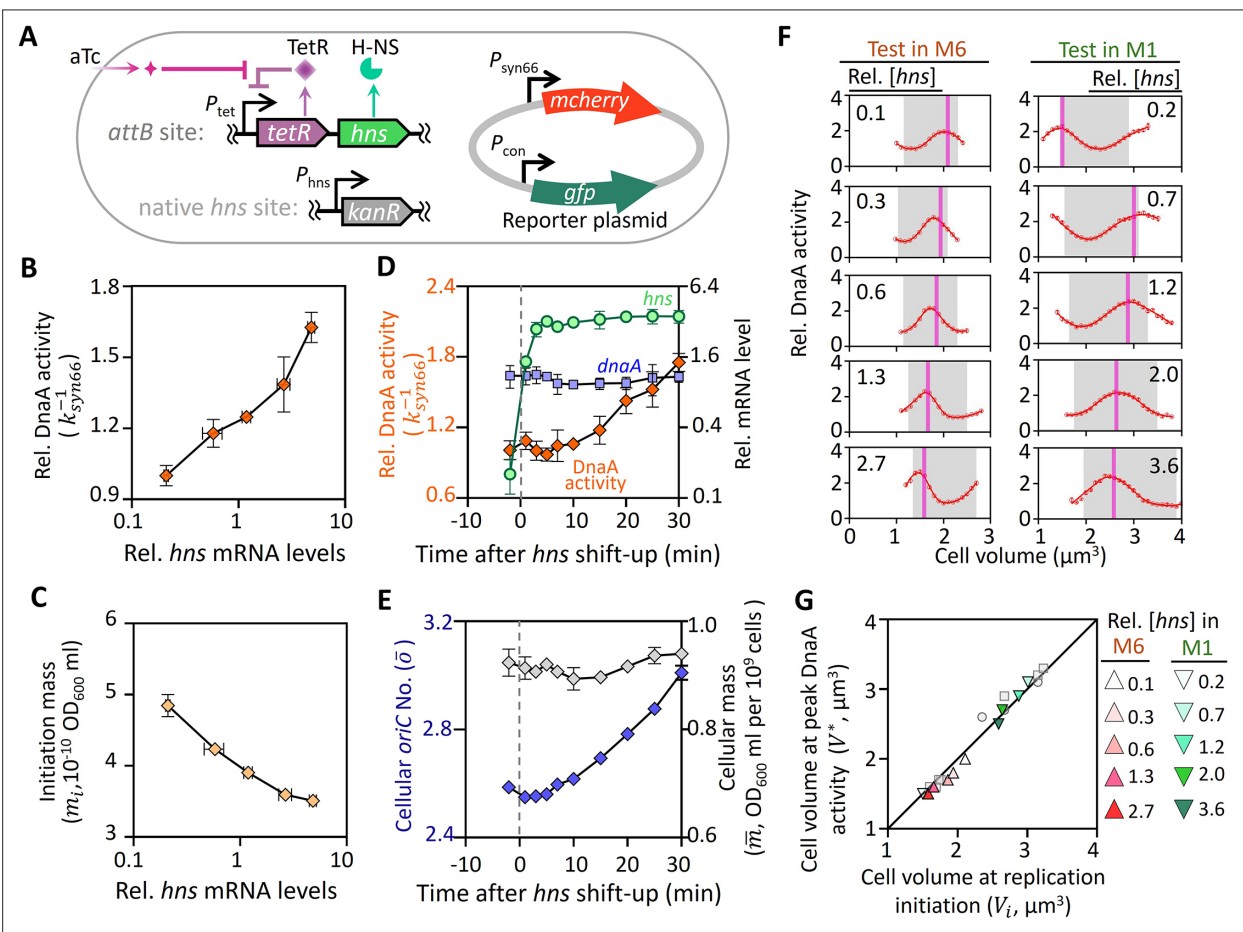

**Figure 7.** Titration of *hns* expression modulates DnaA activity and replication initiation. (**A**) Genetic circuit of the *hns*-titratable strain. Expression of *hns* is controlled by a $P_{tet}$-*tetR* negative feedback loop integrated at the *attB* site, with the native *hns* coding sequence replaced by a kanamycin resistance gene. The plasmid containing $P_{syn66}$-*mcherry* and $P_{con}$-*gfp* expression cassettes was used to assess DnaA activity. DnaA activity (**B**) and initiation mass (**C**) were characterized in M6 medium with varying *hns* expression levels during steady-state cultivation. (**D**) mRNA levels of *hns* (green circles) and *dnaA* (blue squares) relative to wild-type levels, along with DnaA activity (orange rhombus), were measured during *hns* shift-up. *hns* shift-up was induced by the addition of 50 ng·ml⁻¹ aTc at time 0 (dashed line), followed by steady-state cultivation in M6 medium without aTc. (**E**) Dynamics of population-averaged *oriC* number (blue rhombus, left axis) and cellular mass (gray rhombus, right axis) during *hns* shift-up. Data represent mean ± SD from 3 biological replicates (**B–E**). (**F**) Cell cycle-dependent DnaA activity oscillations in *hns*-titratable cells cultivated in M1 and M6 media under varying aTc concentrations. Relative *hns* mRNA levels are indicated for each condition. More than 8000 cells were analyzed for each condition, with at least 150 cells per bin; error bars show the mean ± SEM. (**G**) Correlation between the volume at maximal DnaA activity ($V^*$) and the volume at replication initiation ($V_i$) for *hns*-titratable cells grown in M1 (green down-pointing triangle) and M6 (red up-pointing triangle) media with varying *hns* expression levels. Data for *dnaA*-titratable cells (gray circles) and wild-type *dnaA*-autoregulated cells (*Figure 5C*) are included for comparison.

The online version of this article includes the following source data and figure supplement(s) for figure 7:

**Source data 1.** Source data for *Figure 7* showing the effect of *hns* expression level on DnaA activity and the timing of DNA replication initiation.

**Figure supplement 1.** H-NS promotes the release of DnaA from the *datA* sequence.

**Figure supplement 2.** Basic phenotypic characterization of *hns*-titratable cells harboring a DnaA activity reporting plasmid.

extrusion model. To this end, we constructed a *hns*-titratable strain, incorporating a reporting plasmid to capture changes in DnaA activity across various H-NS expression levels (**Figure 7A**). Under steady-state cultivation, increasing aTc concentrations from 0 to 50 ng·ml⁻¹ resulted in *hns* mRNA levels ranging from 0.25 to 5 times the wild-type levels, without affecting growth rate (**Figure 7—figure supplement 2A and B**). Within this range, DnaA activity increased by approximately 60% (**Figure 7B**), while the initiation mass decreased by 30% (**Figure 7C**; calculated with **Figure 7—figure supplement 2C and D**), in qualitative agreement with the predictions of the extrusion model (**Figure 7—figure supplement 2E and F**).

Since H-NS is a global transcriptional regulator (**Hommais et al., 2001**), we wonder whether its effect on DnaA activity was mediated through *dnaA* transcription. To test this, we measured DnaA activity and *dnaA* mRNA levels during an H-NS shift-up. Upon addition of 50 ng·ml⁻¹ aTc, *hns* levels increased more than 20-fold within a short period, followed by approximately 80% increase in DnaA activity, with no significant change in *dnaA* mRNA levels (**Figure 7D**). Additionally, cellular *oriC* content increased significantly within 30 min, while cell mass remained unchanged (**Figure 7E**). These results suggest that the rapid increase in DnaA activity and accelerated replication initiation induced by H-NS upregulation is not driven by changes in *dnaA* transcription.

Leveraging the *hns*-titratable strain, we re-examined the tight correlation between DnaA activity oscillations and replication initiation. In two different growth media with varying *hns* levels, the peak of DnaA activity fluctuations ($V^*$) shifted in concert with the independently measured volume at replication initiation ($V_i$) (**Figure 7F**). Plotting $V^*$ against $V_i$ revealed the same equivalency observed in wild-type and *dnaA*-titratable cells (**Figure 7G**). Collectively, these results support the notion that DnaA activity oscillations, modulated by extrusion, coordinate DNA replication and biomass growth.

## Discussion

In this study, we proposed chromosome-driven DnaA activity oscillations as a posttranslational 'rheostat' coordinating DNA replication with biomass growth in *E. coli*. By developing a SeqA-independent synthetic promoter reporter ($P_{syn66}$) on a plasmid, we circumvented confounding factors inherent to native systems (**Pountain et al., 2024**), thereby achieving the direct quantification of DnaA activity dynamics. Crucially, these oscillations persist when *dnaA* transcription is fixed, operate independently of SeqA-mediated repression, and peak precisely at replication initiation across all tested growth conditions. This reveals an essential paradigm: bacteria bypass transcriptional control to synchronize genome duplication with growth through regulation of DnaA-chromosome interactions.

Very recently, Iuliani et al. also reported DnaA activity oscillations (**Iuliani et al., 2024**), which is a significant milestone, innovatively using a chromosomal promoter reporter system and microfluidics for single-cell DnaA activity observations. They revealed key relationships between DnaA activity, cell size, and division. Our work, however, differs in multiple ways. Methodologically, we engineered a *dnaA*-titratable strain and synthetic promoters like $P_{syn66}$, enabling precise *dnaA* expression control and cleaner DnaA activity reporting. Our mRNA FISH approach uniquely compares DnaA activity and transcription at the single-cell level. Replication initiation volume scales proportionally with peak DnaA activity volume with a slope of 1.0 ($R^2 = 0.98$, **Figure 7G**), indicating predictive correspondence rather than absolute coincidence. While population-level $V_i$ estimation cannot resolve single-cell stochasticity, the consistent $V^* : V_i$ relationship across 20 conditions suggests DnaA activity thresholds predict initiation timing within physiological error margins. Regarding research questions, while Iuliani et al. focused on DnaA-cell size-division relationships, we addressed the decoupling of replication initiation from *dnaA* expression with our novel extrusion model.

Recently, an integrated 'titration-switch' model has been proposed to improve replication cycle stability across different growth rates (**Berger and Wolde, 2022**; **Fu et al., 2023**). Although this integrated model predicts a relationship between initiation mass and *dnaA* mRNA levels that aligns closely with steady-state experimental data (**Appendix 1—figure 1A**), it fails to account for the additional replication initiation observed after *dnaA* shutdown (**Appendix 1—figure 1B**). By incorporating an extruder into this model, we successfully reproduced the additional replication initiation (**Appendix 1—figure 1C**), thus validating the extrusion concept and aligning with experimental observations. These findings highlight the critical role of extrusion in refining our mechanistic understanding of biomass-DNA coordination.

Mechanistically, we propose that NAPs sense biomass-DNA imbalances and extrude DnaA from chromosomal sites, converting physical displacement into biochemical activation, a spatial control strategy mirroring eukaryotic chromatin regulation. H-NS emerges as a prime extruder candidate given its DNA-condensing ability (*Dame et al., 2000*; *Zimmerman, 2006*) and replication-timing effects (*Figure 7*). This functional multiplicity likely enhances robustness, allowing cells to maintain replication homeostasis despite fluctuating environments, an evolutionary advantage paralleling eukaryotic checkpoint system.

These findings refine bacterial cell cycle control as a dynamic interplay between physical chromosome remodeling and metabolic sensing. By tethering DnaA activation to biomass accumulation through extrusion, cells achieve an important capability: noise suppression via spatial filtering of transcriptional fluctuations. Such mechanistic parsimony, using chromosome structure to both store genetic information and regulate replication timing, may represent a potential organizational strategy. Indeed, parallels to eukaryotic systems are striking: DnaA extrusion resembles ORC licensing through chromatin accessibility (*Fragkos et al., 2015*), while biomass-DNA coupling evokes yeast size-control checkpoints (*Xie et al., 2022*).

While H-NS perturbation supports extrusion mechanism, future work should identify the full extruder interactome and elucidate how metabolic signals modulate their activity. Our synthetic reporter system provides a valuable tool for dissecting the spatial regulation of DNA-binding proteins, with translational potential in antimicrobial development (targeting pathogen-specific extruders) and synthetic biology (engineering tunable replication circuits). More broadly, this work establishes physical chromosome remodeling as a central coordinator of cellular homeostasis, a concept bridging prokaryotic and eukaryotic cell cycle paradigms.

## Materials and methods
### Plasmid and strain construction
Strains used in this study were derived from *E. coli* MG1655 (*Liu et al., 2011*). Information for all strains and plasmids is listed in *Appendix 2—table 1* and *Appendix 2—table 2*, respectively.

### Plasmids for reporting DnaA activity
Plasmids expressing GFP, mCherry, or LacZ under synthetic promoters were constructed based on the pPT plasmid backbone (*Zong et al., 2017*). The *lacZ* amplified from the MG1655 genome and mCherry gene amplified from pMD19-*hupA-mcherry* plasmid were assembled into the BspQI/BsrGI-digested pPT backbone via the One Step Cloning Kit (Vazyme, C112), to generate the pPT-lacZ and pPT-RFP plasmids, respectively. Synthetic promoters, generated by annealing the primer pairs listed in *Appendix 2—table 3*, were ligated directly into the BsaI-digested pPT, pPT-RFP, or pPT-lacZ plasmid as required. The native *dnaA* promoter ($P_{native}$) amplified from the MG1655 genome was digested with BsaI and ligated into the BsaI-digested pPT plasmid. The $P_{con}$-*gfp* cassette amplified from the $P_{con}$-GFP plasmid was assembled into the SpeI-digested $P_{syn66}$-RFP plasmid to generate the $P_{sny66}$-$P_{con}$-FPs plasmid. Primer information for amplifying DNA fragments is detailed in *Appendix 2—table 4*.

### Plasmids for strain construction
For constructing the plasmid pMD19-Rhns, pMD19-RseqA, and pMD19-RdnaA, the *hns*, *seqA*, and *dnaA* genes were amplified from MG1655 genome, digested, and ligated into the same enzyme-digested pMD19-tetR plasmid (*Zheng et al., 2016*). For constructing the plasmid p15A-RdnaA, the T3-$P_{tet}$-*tetR-dnaA* cassette, *kan*[r] gene, and p15A-T1 fragment were amplified from plasmid pMD19-RdnaA, pEcCas (*Li et al., 2021*), and PZA31-$P_{tet}$-M2-GFP (*Liu et al., 2019*), respectively, and then assembled via MultiS One Step Cloning Kit (Vazyme, C113). The plasmid P_CRidnaA1 was obtained by ligating the annealed primer pairs to the BsaI-digested CPP00458 plasmid for the *dnaA* targeting. The chloramphenicol resistance gene *Cm*[r] amplified from the PZA31-$P_{tet}$-M2-GFP plasmid replaced the *kan*[r] gene in the pEcCas plasmid via $\lambda$ Red recombination method, resulting in the CmPcas plasmid. Primer information is detailed in *Appendix 2—table 4*.

## Strain construction

The *seqA*- and *hns*-titratable strains were constructed as previously described (*Zheng et al., 2016*). Briefly, after transforming the pSIM5 plasmid into MG1655 cells, the $P_{tet}$-*tetR*-driven *seqA* or *hns* cassette was inserted at the *attB* site on the genome, and then the native gene was replaced with the *kan*^r gene amplified from plkml plasmid.

For the construction of the *dnaA*-titratable strain, we first inserted the $P_{tet}$-*tetR cassette* into the *intS* site via CRISPR/Cas9-$\lambda$ Red genome editing system (*Jiang et al., 2015*) by targeting 20-nucleotide sequence (tcttcctgcagaccagatcc). Then, applying the $\lambda$ Red recombination systems in the CmPcas plasmid, the *bla*:$P_{tet}$-*dnaA* cassette amplified from the pMD19-RdnaA plasmid via overlap PCR was inserted between the *yid*A and *yid*X genes, and the native $P_{native}$-*dnaA* was replaced with the $T_3$-Kan-$T_1$ cassette amplified from the p15A-RdnaA plasmid.

In order to delete the native *lacZ* gene in the genome of the RdnaA1, Rhns1, and MG1655 cells, the homologous recombination fragment amplified from the genome of the CL1 strain was used to knock out the native *lac* operon via CRISPR/Cas9-$\lambda$ Red editing system (*Jiang et al., 2015*) by targeting 20-nucleotide sequence (cttccggctcgtatgttgtg).

For the construction of DnaA shutdown strain (CRidnaA1), the $P_{J23119}$-*sgRNA*$_{dnaA}$:$P_{J23100}$-*tetR*:$P_{tet}$-*dUn1Cas12f1* cassette was amplified from the P_CRidnaA1 plasmid and flanked with homologous sequences to the *asn*A and *via*A gene. This cassette was then inserted into the MG1655 genome using the CRISPR/Cas9-$\lambda$ Red editing system, targeting the 19-nucleotide sequence (aagccgcctgct cagacgc). The CPP00458 plasmid, kindly provided by the Xiongfei Fu lab, contains the Un1Cas12f1 variant, which carries mutations D143R, T147R, D326A, and K330R that abolish its DNA-cleaving activity. However, it retains the ability to bind DNA in an sgRNA-guided manner, thereby inhibiting the expression of the target gene.

All primers for the construction of DNA fragments are detailed in *Appendix 2—table 4*.

## Growth conditions

Cells for strain construction were grown in Luria-Bertani (LB) medium with appropriate antibiotics: ampicillin (100 μg·ml⁻¹), kanamycin (50 μg·ml⁻¹), spectinomycin (50 μg·ml⁻¹), chloramphenicol (25 μg·ml⁻¹), based on plasmid presence. For titratable strains, 10 ng·ml⁻¹ aTc was added during strain construction and seed culture preparation. Seed cultures were initiated by inoculating three to six individual colonies from an LB agar plate into 14 ml round-bottom test tubes containing 2 ml of the desired medium for overnight growth.

For physiological phenotypic characterization, all cells were cultivated at 37°C with shaking at 150 rpm in a water bath shaker. The *dnaA*-, *seqA*-, and *hns*-titratable cells were characterized under varying aTc concentrations in rich defined medium supplemented with glycerol (M6). *hns*-titratable cells were also characterized in rich defined medium supplemented with glucose (M1). When strains contained reporter plasmid, 25 μg·ml⁻¹ chloramphenicol was added. The autoregulated-*dnaA* strain (MGCL1) with a reporter plasmid was tested in seven different media: M1 (MOPS+EZ+AUCG+Glucose), M2 (MOPS+EZ+Glucose), M3 (MOPS+EZ+ AUCG+Gluconate), M6 (MOPS+EZ+ AUCG+Glycerol), M12 (MOPS+EZ+Glycerol), M13 (MOPS+CAA+Glycerol), and M18 (MOPS+Glucose). Detailed medium compositions were provided previously (*Zheng et al., 2020*).

Steady-state growth was established as previously described (*Zheng et al., 2020*) by culturing cells for ~10 generations in exponential growth phase with ~15-fold dilution at $OD_{600}$ ~0.2, using the same medium supplemented with antibiotic or aTc as required. $OD_{600}$ measurements were taken at consecutive time points using a spectrometer (Genesys 10s, Thermo Fisher Scientific), and the growth rate (or doubling time) $\lambda$ was calculated by fitting the data to an exponential growth curve. Samples for determination of population-averaged cell mass ($\overline{m}$) and cellular origin number ($\overline{o}$), quantification of mRNA levels, and mRNA FISH were collected when $OD_{600}$ reached ~0.2.

## Measurements of population-averaged cellular mass ($\overline{m}$), cellular *oriC* number ($\overline{o}$), the initiation to division period (C+D), and initiation mass ($m_i$)

The population-averaged cell mass ($\overline{m}$), cellular *oriC* number ($\overline{o}$), and initiation mass ($m_i$) were calculated as described previously (*Zheng et al., 2020*). Briefly, $\overline{m}$ was determined by dividing the $OD_{600}$ by the cell concentration ($OD_{600}$ ml per $10^9$ cells). For cell counting, samples were collected in precooled

cell count buffer (0.9% NaCl with 0.12% formaldehyde, filtered with a 0.22 μm filter) and stained with 1 μg·ml⁻¹ DAPI in the same buffer. Cell number concentration was measured by counting the DAPI-stained particles using flow cytometry equipped with 405 nm laser (CytoFLEX, Beckman Coulter Life Sciences). The $\bar{o}$ was calculated based on the classical run-out experiments (*Zheng et al., 2020*). In brief, cell suspensions were treated with 500 μg·ml⁻¹ rifampicin and 30 μg·ml⁻¹ cephalexin to inhibit new initiation and cell division, respectively, and maintained for two to three mass doublings to complete ongoing replication. Cells were fixed with 70% precooled absolute ethanol and treated with stain buffer (20 mM Tris-HCl, pH 8.0, 130 mM NaCl, 10 ng·ml⁻¹ DAPI). Using the same flow cytometry, we measured the cellular origin distribution based on DAPI signaling, from which $\bar{o}$ was calculated. C+D was calculated based on the equation: C+D = ln($\bar{o}$)/ $\lambda$ . $m_i$ was calculated based on the equation: $m_i = \frac{\bar{m}}{\bar{o}} \times \frac{1}{ln2}$ (*Bremer et al., 1979*; *Si et al., 2017*; *Zheng et al., 2016*).

## Screening of the synthetic promoter library

After transforming the reporter plasmid into *dnaA*-titratable cells, the seed culture was cultivated in M6 medium with 10 ng·ml⁻¹ aTc, then diluted 1:180 in fresh medium containing either 0.5 or 50 ng·ml⁻¹ aTc and cultivated at 37°C and 800 r.p.m. in a digital thermo-shaker (AOSHENG) using flat-bottom 96-well plates (Corning). Following approximately 4 hr of pre-cultivation, the cultures were diluted 1:180 in fresh medium with the same aTc concentration. The OD$_{600}$ and GFP fluorescence intensity were continuously monitored during cell growth using a Synergy H1 microplate reader (BioTek). The nonfluorescent strain (MG1655) cultured in 180 μl of M6 medium for each experiment was used to subtract the background fluorescence. For OD$_{600}$ between 0.1 and 0.2, the repression fold change was calculated by dividing the mean fluorescence intensity per OD$_{600}$ of the cells grown with 0.5 ng·ml⁻¹ aTc induction by the corresponding intensity for cells grown with 50 ng·ml⁻¹ aTc induction.

## Quantitative real-time PCR

Quantitative real-time PCR was performed using a protocol similar to previously described (*Zheng et al., 2016*). Briefly, RNA was stabilized using Bacteria Protect Reagent (QIAGEN) and extracted using RNeasy Mini Kit (QIAGEN). 500 ng RNA was reverse-transcribed (TAKARA), and the resulting cDNA samples were diluted 1:25 with PCR-grade water. Quantitative real-time PCR was carried out on a qTower3 (Analytikjena) or CFX Connect system (Bio-Rad), using the following program: 30 s at 95°C, followed by 40 cycles of denaturation (5 s at 95°C), annealing, and elongation (30 s at 58°C). Primer sequences were listed in *Appendix 2—table 4*. The *rpoA* gene was used as the reference gene.

## *lacZ* and *dnaA* mRNA FISH

FISH for *lacZ* or *dnaA* mRNA was performed using a method similar to that previously described (*Skinner et al., 2013*). Briefly, antisense DNA oligo probes were designed against *lacZ* or *dnaA* and synthesized with TAMRA labeling at the 3′ end. The sequences of probes targeting *lacZ* or *dnaA* are either as previously described (*Skinner et al., 2013*) or listed in *Appendix 2—table 5*. 27 ml steady-state growth culture (OD$_{600}$~0.2) was fixed by directly adding 3 ml of 37% (vol/vol) formaldehyde, followed by incubation for 30 min at room temperature with gentle mixing using a nutator. The fixed cells were collected by centrifugation at 600 × *g* for 8 min and washed twice with 1 ml RNase-free PBS. The cell pellet was resuspended in 300 μl DEPC water, and 700 μl precooled absolute ethanol was added to permeabilize the cells for 1 hr at room temperature in the nutator. After permeabilization, the cells were divided into two equal parts for characterization of *lacZ* or *dnaA* mRNA as needed. Hybridization was performed as described previously (*Skinner et al., 2013*), with the stringency of the hybridization and wash solutions adjusted by changing the concentration of formamide, 30% for *dnaA* probes and 40% for *lacZ* probes. After hybridization, the cells were washed with the appropriate wash solution and prepared for imaging.

## Cell imaging and cell size parameters measurement

For imaging, 3 μl of cell suspension was pipetted onto the center of a 24×60 mm² coverslip. A thin 1.5% agarose pad (prepared with 1x PBS) was then placed on top of the cell suspension droplet. Fluorescent and phase images were acquired using an inverted microscope (IX-83, Olympus) equipped with a 100x oil objective (Olympus), an automated xy-stage (ASI, MS2000), and a sCMOS camera (Prime BSI, photometrics). Fluorescence excitation was performed as previously described (*Xia et al.,*

*2022*). Solid-state lasers (Coherent OBIS: 405-100 LX, 561-100 LS) and Semrock emission bandpass filters (445/40 nm for DAPI and 598/25 nm for RFP) were used for excitation and collecting fluorescent signals.

Images were aligned with the phase-contrast channel by selecting at least six pairs of common control points and applying a geometric transformation using the MATLAB imwarp function. This allowed for adjustment of the cell masks in the RFP or DAPI channel. A customized ImageJ-based image-processing package, MicrobeJ (v5.13l₁) plug-in for Fiji (*Ducret et al., 2016*), was used to contour cells and calculate cell size parameters and fluorescence intensity, including the mean cell width, cell length, and cell area from phase-contrast images. The cell volume was calculated using a cylinder-plus-two-hemispheres geometry. The total RFP intensity per cell volume, corrected for background subtraction, was then used to estimate the mRNA concentration of *lacZ* or *dnaA*.

## Characterization of cell cycle-dependent fluctuations in DnaA activity

Data filtering was first performed based on the intensity of the DAPI fluorescence signal. To eliminate non-cell regions resulting from incorrect segmentation of the phase-contrast images, only DAPI-positive cells were included in the analysis. To further minimize the potential impact of variability in fixation, permeabilization, and wash efficiencies, which could affect the TAMRA fluorescence signal (representing the mRNA FISH signal), only cells with mean DAPI signals within the 95% confidence intervals were included in the subsequent analysis.

The concentration of the *lacZ* mRNA ($[mZ]$) can be expressed as $[mZ] = J/\delta_Z$, where $\delta_Z$ is the degradation rate and $J$ is the mRNA synthesis flux. $J$ is in turn given by the promoter on-rate ($k$), the RNAP concentration ($[RNAP]$), and the concentration of the *lacZ* gene ($[G_Z]$). Assuming that the factors $\delta_Z$, [RNAP], and $[G_Z]$ are the same between the two strains with $P_{syn66}$ and $P_{con}$ driving *lacZ*, then the ratio of the mRNA concentration obtained at each point of the cell cycle gives the ratio of the promoter on-rates, $k_{Psyn66}$ and $k_{Pcon}$. Since a constitutive promoter is nonresponsive to DnaA activity, $k_{con}$ is independent of cell cycle and we can obtain DnaA activity-regulated promoter activity as $k_{syn66} = \frac{[mZ]\,(\mathrm{P_{syn66}})}{[mZ]\,(\mathrm{P_{con}})}$, then the DnaA activity is denoted as $k_{syn66}^{-1}$.

The concentration of the *lacZ* mRNA was binned based on cell volume, with a bin size of 0.1 µm³, and was used to determine $[mZ]_{(Psyn66)}$ and $[mZ]_{(Pcon)}$. The relation between $[mZ]_{(Pcon)}$ and cell volume was smoothed using the MATLAB smooth function, and from this, $k_{syn66}$ was deduced.

Based on the DnaA activity oscillations, the cell volume where DnaA activity reaches maximum ($V^*$) was calculated during a representative birth-to-division cell cycle. The cell volume distribution was used to calculate the number of cells in each volume-doubling range. The range containing the highest cells was considered the representative birth-to-division cell cycle and was shaded in gray (*Figure 5—figure supplement 1D*). From this, $V^*$ was obtained.

## Calculation of cell volume at the time of replication initiation ($V_i$)

Cell volume at the time of replication initiation ($V_i$) was calculated based on population-averaged cell volume ($\bar{V}$) and the population-averaged cellular *oriC* number ($\bar{o}$). The equation used for this calculation is as follows:

$$V_i = v_i \times N_{oriC}^{ini.} = \frac{\bar{V}}{\bar{o}} \times \frac{1}{ln2} \times 2^{\lfloor log_2^{\bar{o}} \rfloor}$$

where $v_i$ is the initiation volume by definition, i.e., cell volume per *oriC* at the time of replication initiation (*Bremer et al., 1979*; *Si et al., 2017*; *Zheng et al., 2016*). $\bar{V}$ was obtained by analyzing the phase-contrast images, $\bar{o}$ was obtained by run-out experiments (*Zheng et al., 2020*). $N_{oriC}^{ini.}$ is the cellular *oriC* number at the time of replication initiation. $\lfloor log_2 \bar{o} \rfloor$ denotes the floor function of $log_2 \bar{o}$.

### DnaA shutdown experiment

To shut down DnaA expression, steady-state growth of the CRidnaA1 cells was first established after ~10 generations of exponential growth in M1 medium. At OD₆₀₀ between 0.15 and 0.2, 24 ml of cell culture was transferred to 12 ml of pre-warmed M1 medium containing 150 ng·ml⁻¹ aTc to induce dUn1Cas12f1 expression at time 0. Under the guidance of sgRNA_dnaA, the transcription of the *dnaA* gene was inhibited. Samples were collected at various time points to determine population-averaged cellular mass ($\bar{o}$), cellular *oriC* number ($\bar{o}$), and to quantify mRNA levels. When the OD₆₀₀ reached

around 0.2, the cultures were diluted about 2-fold with pre-warmed M1 medium containing 50 ng·ml⁻¹ aTc to maintain adequate nutrition. The steady-state growing MG1655 cells in M1 medium were also collected for quantification of wild-type *dnaA* mRNA levels. Three independent experiments were conducted.

## H-NS shift-up experiment

The DnaA activity reporter plasmid $P_{syn66}$-$P_{con}$-FPs was transformed into the *hns*-titratable strain (Rhns2), and cells were cultivated overnight in M6 medium containing 10 ng·ml⁻¹ aTc and 25 μg·ml⁻¹ chloramphenicol for seed culture. Seed cultures were washed and grown in the same medium without aTc for ~10 generations of exponential growth. At an $OD_{600}$ of ~0.2, 40 ml of cell culture was transferred to 20 ml of pre-warmed same medium containing 150 ng·ml⁻¹ aTc to induce H-NS expression at time 0. Samples were collected at time points –2, 1, 3, 5, 7, 10, 15, 20, 25, and 30 min for the determination of population-averaged cellular mass ($\bar{m}$), cellular *oriC* number ($\bar{o}$), and quantification of mRNA levels. Steady-state growing MG1655 in M6 medium was also collected for *hns* mRNA quantification. Three independent experiments were conducted.

## Electrophoretic mobility shift assay

The genes encoding *dnaA* and *hns* from the MG1655 genome were individually amplified and cloned into the expression vector pET-28a using primers listed in **Appendix 2—table 4**. *E. coli* BL21 (DE3) carrying pET-28a-DnaA or pET-28a-H-NS was used for overproduction of $His_6$-DnaA or $His_6$-H-NS. For each protein purification, a 1 l culture was induced by the addition of 1 mM IPTG at $OD_{600}$=0.4. After 1.5 hr of incubation, cells were harvested by centrifugation (5000 rpm for 20 min at 4°C). The pellet was resuspended in 50 ml of buffer A (50 mM Tris-HCl, pH 7.0, 500 mM NaCl, 0.5 mM PMSF, 5% glycerol). Cell resuspension was lysed on ice for 30 min (3 s on, 7 s off) using a sonicator at 4°C, and cell debris was removed by centrifugation (14,000 rpm for 60 min at 4°C). Subsequently, the protein was purified from the soluble supernatant by $Ni^{2+}$-affinity chromatography (HisTrap HP Columns; GE Healthcare) with a ÄKTA Pure Protein Purification System. Proteins were collected using gradient elution with buffer B (20 mM Tris-HCl, pH 7.5, 500 mM NaCl, 400 mM imidazole, 5% glycerol). Eluted proteins were further purified and concentrated by ultrafiltration using Amicon Ultra 10K (for H-NS) and 30K (for DnaA) filters (Millipore). Finally, after overnight dialysis (0.025 μm Membrane Filter; MF-Millipore) at 4°C, the protein was stored in buffer C (45 mM HEPES-KOH, pH 7.6, 600 mM potassium glutamate, 1 mM DTT, 10 mM magnesium acetate, 0.5 mM EDTA, 20% glycerol). Judged by SDS-PAGE with Coomassie Brilliant Blue staining, the purity of each protein was >90%.

For the experiments where H-NS acts on the *datA* bound by DnaA, the purified DnaA, *datA*, and ATP were first incubated in buffer D (20 mM Tris-HCl, pH 7.5, 150 mM potassium glutamate, 10 mM magnesium acetate, 8 mM DTT) at 20°C for 10 min. Then, the H-NS was added to buffer D (15 μl) which contained DnaA-*datA* complexes and incubated for an additional 10 min at 20°C. The final complexes were separated by electrophoresis in 2% (wt/vol) agarose gel at 80 V for 70 min in Tris-Borate buffer, followed by staining with SYBR Safe (Invitrogen).

## Titration model

In the titration model, we posit that the cell mass grows exponentially according to the differential equation $\frac{dV}{dt} = \lambda V$, where $V$ represents the volume of the cell and $\lambda = 1.54\,h^{-1}$ (corresponding to doubling time of 27 min) is the growth rate. Concurrently, the number of total DnaA protein ($A$) increases at a rate proportional to the cell's mass, described by $\frac{dA}{dt} = \alpha_A V$, with $\alpha_A = 300\,h^{-1}\mu m^{-3}$ denoting the synthesis rate per unit volume. The chromosome's structure is abstracted as a vector representing the copy numbers of uniformly distributed sites along its length, and the number of DnaA-boxes ($A_b$) is quantified by summing the boxes number at their respective locations. In this model, 300 DnaA-boxes are evenly distributed on the chromosome while an additional 100 DnaA-boxes are located near the *oriC*. This arrangement approximates a head-weighted distribution of DnaA-boxes and helps prevent over-initiation (**Hansen et al., 1991**). Binding of DnaA proteins to these DnaA-boxes allows us to calculate the concentration of free DnaA as $\left[A_f\right] = \left(A - A_b\right)/V$, under condition of $\left(A - A_b\right) \geq 0$. DNA replication commences once the concentration of free DnaA exceeds a critical threshold $\left[A_f^c\right] = 10\,\mu m^{-3}$: $\left[A_f\right] > \left[A_f^c\right]$. 10 min sequestration time was set after replication initiation to mimic the sequestration time (**Campbell and Kleckner, 1990**).

To explore the dynamics of the *oriC* number following the cessation of *dnaA* expression, we can set the synthesis rate to be zero $\alpha_A = 0$. To simulate the dependence of initiation mass on *dnaA* expression, $\alpha_A$ varies from $60\ h^{-1}\mu m^{-3}$ to $3840\ h^{-1}\mu m^{-3}$.

## Extrusion model

The extrusion model was formulated following the titration model, with an additional extruder introduced as a protein with synthesis rate $\alpha_H = 180\ h^{-1}\mu m^{-3}$ and dynamics $dH/dt = \alpha_H V$. The additional extruder competes the DnaA-boxes with DnaA proteins, with higher affinity to DnaA-boxes. So that it can displace DnaA from the DnaA-boxes, leading to a revised calculation for free DnaA concentration as $[A_f] = (A - (A_b - H))/V$, under condition of $(A_b - H) \geq 0$ and $(A - (A_b - H)) \geq 0$. To get stochastic modeling for the adder phenomenon, we introduced white noises on cell mass growth rate and the DnaA synthesis rate at each replication cycle: $\lambda = \lambda_0 + \eta\lambda_0$, $\alpha_A = \alpha_0 + \eta\alpha_0$, where $\lambda_0 = 0.66\ h^{-1}, \alpha_0 = 300\ h^{-1}\mu m^{-3}$, and $\eta = 0.1$ representing white noise. This extrusion model exhibits consistent performance under various parameter ranges.

## DnaA-ATP/DnaA-ADP switch model

The switch model was simulated following the idea of the LDDR model proposed by **Berger and Wolde, 2022**. The formulations of the DnaA-ATP fraction dynamics are:

$$\frac{df}{dt} = \left(\frac{\alpha_l}{[D]_T}V + \frac{\alpha_{d1}}{[D]_T}N_{d1} + \frac{\alpha_{d1}}{[D]_T}N_{d2}\right)\frac{1-f}{\frac{K_D}{[D]_T} + 1 - f} - \left(\frac{\beta_{datA}}{[D]_T} + \frac{\beta_{rida}}{[D]_T}\right)N_o\frac{f}{\frac{K_D}{[D]_T} + f} + \frac{\alpha_A}{[D]_T}\lambda\left(1 - f\right)$$

In this equation $[D]_T = 400\ \mu m^{-3}$ is the concentration of total DnaA proteins in a cell, and $\alpha_l = 750\ h^{-1}$ is the activation rate of DnaA-ATP by lipid; $\alpha_{d1} = 100\ h^{-1}$, $\alpha_{d2} = 643\ h^{-1}$ are the activation rates of DnaA-ATP by the DARS1 and DARS2 sites; $\beta_{datA} = 600\ h^{-1}$, $\beta_{rida} = 500\ h^{-1}$ are the deactivation rates of DnaA-ATP by *datA* and RIDA; $K_D = 50\ \mu m^{-3}$ the dissociation constant of DnaA activation and deactivation; $N_{d1}, N_{d2}, N_o$ are copy numbers of DARS1, DARS2, and *oriC* sites extracted dynamically from the DNA copy number vector; $\alpha_A = 300\ h^{-1}\mu m^{-3}$ is the DnaA expression level. The DNA replication initiation was determined when $f$ reaches a threshold $f_C = 0.75$ (**Berger and Wolde, 2022**). To simulate the *dnaA* shutdown, we simply deleted the term of *dnaA* synthesis ($\lambda(1 - f)$).

## Titration-switch model and titration-switch-extrusion model

In the integrated model combining titration and switch mechanisms, replication initiation is triggered when the concentration of free DnaA-ATP $[A^f_{ATP}]$ exceeds a threshold $[A^{fc}_{ATP}]$. The concentration of free DnaA-ATP is titrated by the DnaA-boxes on the chromosome as in the titration model, with $[A^f_{ATP}] = (A_{ATP} - A_b)/V$, under condition of $(A_{ATP} - A_b) \geq 0$. The parameters related to the titration dynamics remain consistent with those described in earlier sections of the titration model.

The dynamics of total DnaA-ATP ($A_{ATP}$) is governed by both expression and conversion processes as described in **Berger and Wolde, 2022**. According to this framework, the lipid and DARS systems facilitate the conversion of DnaA-ADP to DnaA-ATP at constant rates, while the *datA* site and RIDA systems also play a role in converting DnaA-ATP to DnaA-ADP, again at fixed rates. Additionally, the synthesis of new DnaA proteins provides another source of DnaA-ATP.

To be specific, DNA replication initiation was set when the concentration of free DnaA-ATP, $[A^{ATP}_f] = (A^{ATP} - A_b)/V$ exceeds a threshold $[A^c_f]$. The dynamics of total DnaA number in a cell $A$ followed $\frac{dA}{dt} = \frac{\phi_{P0}\lambda V}{1 + \left([A^{ATP}_f]/K^P_D\right)^n}$, and the dynamics of total DnaA-ATP number $A^{ATP}$ followed:

$$\frac{dA^{ATP}}{dt} = (\alpha_l V + \alpha_{d1}N_{d1} + \alpha_{d2}N_{d2})\frac{[A^{ADP}]}{K_D + [A^{ADP}]} - (\beta_{datA} + \beta_{rida})N_o\frac{[A^{ATP}_f]}{K_D + [A^{ATP}_f]} + \frac{\phi_{P0}\lambda V}{1 + \left([A^{ATP}_f]/K^P_D\right)^n}$$

where the three terms on the right-hand side represent activation and deactivation of DnaA-ATP and new synthesis of DnaA-ATP. In these equations, $[A^{ADP}] = \frac{(A - A^{ATP})}{V}$ is the concentration of DnaA-ADP; $[D_T] = 400\ \mu m^{-3}$ is the total DnaA concentration; $K^P_D = 400\ \mu m^{-3}$ is the dissociation constant of DnaA promoter; $\phi_{p0}$ is the DnaA protein ratio relative to all proteins, which is proportional to $\alpha_A$ and ranges

from 0.002% to 0.1%; $n = 5$ is the cooperativity of *dnaA* expression. The same activation/deactivation rates and dissociation constants of DnaA ($\alpha_l, \alpha_{d1}, \alpha_{d2}, \beta_{datA}, \beta_{rida}, K_D$) are used as in the switch model; $N_{d1}, N_{d2}, N_o$ are copy numbers of DARS1, DARS2, and *oriC* sites extracted dynamically from the DNA copy number vector. To get a good fit of the initiation mass-*dnaA* expression relation, high threshold of free DnaA-ATP concentration was applied, $\left[A_{ATP}^{fc}\right] = 200 \, \mu m^{-3}$.

To simulate the titration-switch-extrusion model, we incorporated the extruder as described in the extrusion model, where its amount was given by $H = \alpha_H V$. The concentration of free DnaA-ATP was then calculated as $\left[A_{ATP}^{f}\right] = \left(A_{ATP} - (A_b - H)\right)/V$, under condition of $(A_b - H) \geq 0$ and $(A_{ATP} - (A_b - H)) \geq 0$. To explore the best fit of continuous DNA replication rounds following *dnaA* shutdown, $\alpha_H = 550 \, h^{-1}\mu m^{-3}$ is used.

All parameters are summarized in *Appendix 1—table 1*.

## Acknowledgements

We thank numerous colleagues for discussions. We sincerely thank Dr. Aiguo Xia for his help with microscopy, and Dr. Sheng Yang for providing valuable suggestions on CRISPR-Cas9 system. Special thanks to Drs. Pan Chu and Yeqing Zong for kindly gifting us the CPP00458 and pPT plasmids. This research was financially supported by the National Natural Science Foundation of China (32025022, 32230062), Strategic Priority Research Program of the Chinese Academy of Sciences (XDB0480000), Joint NSFC-ISF Research Grant (32061143021) to CL, US-Israel Binational Science Foundation (2017004) and Israel Science Foundation-NSFC Joint Research Program (3320/20) to AZ National Natural Science Foundation of China (32170042), National Key R&D Program of China (2024YFA0916403), Youth Innovation Promotion Association CAS (number 2022369) to HZ, National Key R&D Program of China (2021YFA0910703) to YB, National Centre for Engineering Biology (NRF-MSG-2023–0003), and National University of Singapore (NUHSRO/2024/064/NUSMed/05/SynCTI2.0) to MWC.

## Additional information

### Funding

| Funder | Grant reference number | Author |
|---|---|---|
| National Natural Science Foundation of China | 32025022 | Chenli Liu |
| National Natural Science Foundation of China | 32230062 | Chenli Liu |
| Chinese Academy of Sciences | Strategic Priority Research Program of the Chinese Academy of Sciences XDB0480000 | Chenli Liu |
| National Natural Science Foundation of China | 32170042 | Hai Zheng |
| National Key Research and Development Program of China | 2024YFA0916403 | Hai Zheng |
| Chinese Academy of Sciences | Youth Innovation Promotion Association of the Chinese Academy of Sciences 2022369 | Hai Zheng |
| National Key Research and Development Program of China | 2021YFA0910703 | Yang Bai |
| National University of Singapore | NUHSRO/2024/064/ NUSMed/05/ SynCTI2.0 | Matthew Chang |

| Funder | Grant reference number | Author |
|---|---|---|
| United States-Israel Binational Science Foundation | 2017004 | Arieh Zaritsky |
| National Centre for Engineering Biology | NRF-MSG-2023-0003 | Matthew Chang |
| Israel Science Foundation | Joint NSFC-ISF Research Grant 32061143021 | Chenli Liu |
| National Natural Science Foundation of China | Joint NSFC-ISF Research Grant 32061143021 | Chenli Liu |
| Israel Science Foundation | Israel Science Foundation-NSFC joint research program 3320/20 | Arieh Zaritsky |
| National Natural Science Foundation of China | Israel Science Foundation-NSFC joint research program 3320/20 | Arieh Zaritsky |

The funders had no role in study design, data collection and interpretation, or the decision to submit the work for publication.

## Author contributions

Dengjin Li, Data curation, Software, Formal analysis, Validation, Investigation, Visualization, Methodology, Writing – original draft, Writing – review and editing; Hai Zheng, Data curation, Formal analysis, Funding acquisition, Validation, Investigation, Visualization, Methodology, Writing – original draft, Writing – review and editing; Yang Bai, Data curation, Software, Formal analysis, Funding acquisition, Validation, Visualization, Methodology, Writing – original draft, Writing – review and editing; Zheng Zhang, Software, Formal analysis, Investigation, Writing – review and editing; Hao Cheng, Formal analysis, Validation, Investigation, Methodology, Writing – review and editing; Xiongliang Huang, Ting Wei, Formal analysis, Validation, Investigation, Writing – review and editing; Matthew Chang, Funding acquisition, Validation, Writing – review and editing; Arieh Zaritsky, Funding acquisition, Validation, Investigation, Writing – review and editing; Terence Hwa, Conceptualization, Formal analysis, Validation, Methodology, Writing – review and editing; Chenli Liu, Conceptualization, Resources, Formal analysis, Supervision, Funding acquisition, Investigation, Visualization, Writing – original draft, Project administration, Writing – review and editing

## Author ORCIDs

Dengjin Li ⓘ https://orcid.org/0009-0001-4148-5476
Hai Zheng ⓘ https://orcid.org/0000-0002-7561-2269
Yang Bai ⓘ https://orcid.org/0000-0001-9976-2686
Zheng Zhang ⓘ http://orcid.org/0000-0002-7824-211X
Ting Wei ⓘ http://orcid.org/0000-0003-2926-5849
Terence Hwa ⓘ https://orcid.org/0000-0003-1837-6842
Chenli Liu ⓘ https://orcid.org/0000-0003-3029-7207

Reviewer #1 (Public review): https://doi.org/10.7554/eLife.107214.3.sa1
Reviewer #2 (Public review): https://doi.org/10.7554/eLife.107214.3.sa2
Author response https://doi.org/10.7554/eLife.107214.3.sa3

# Additional files

## Supplementary files
MDAR checklist

## Data availability
The cellular parameters obtained after mRNA FISH treatment have been deposited on Dryad (DOI: https://doi.org/10.5061/dryad.bvq83bkp4). Source data files have been provided for Figures 2-7.

Simulation data can be generated using the custom-made code and the parameter sets provided. The code is publicly available on GitHub (copy archived at *Bai, 2025*).

The following dataset was generated:

| Author(s) | Year | Dataset title | Dataset URL | Database and Identifier |
|---|---|---|---|---|
| Li D, Zheng H, Bai Y, Zhang Z, Cheng H, Huang X, Wei T, Chang M, Zaritsky A, Hwa T, Liu C | 2025 | Extrusion-modulated DnaA activity oscillations coordinate DNA replication with biomass growth | https://doi.org/10.5061/dryad.bvq83bkp4 | Dryad Digital Repository, 10.5061/dryad.bvq83bkp4 |

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

# Appendix 1

## Supplementary information for model simulation, model parameters, and model interpretation

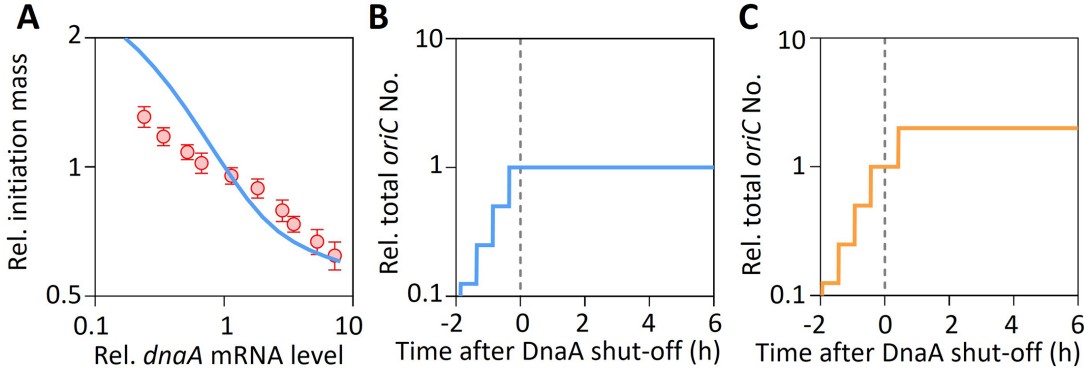

**Appendix 1—figure 1.** Predictions of the titration-switch model and titration-switch-extrusion model. (**A**) Comparison of the titration-switch model predictions and experimental data for the relation between relative initiation mass and *dnaA* mRNA levels. The relative initiation mass was calculated as mass/*oriC* averaged over more than 100 cycles after steady DNA replication initiation was established, while the relative *dnaA* mRNA was achieved by setting various *dnaA* expression rates $\alpha_A$. Dynamics of total *oriC* number during DnaA shutdown predicted by titration-switch model (**B**) and titration-switch-extrusion model (**C**), normalized to the value at DnaA shutdown (dash line).

**Appendix 1—table 1.** Parameters used in models.

| Parameter | Description | Values | Source |
|---|---|---|---|
| **Titration model and extrusion model** | | | |
| $\lambda$ | Biomass growth rate | $1.54\,h^{-1}$ | This study (if not specifically defined) |
| $\alpha_A$ | DnaA synthesis rate | $300\,h^{-1}\mu m^{-3}$ | Fitted in this study |
| $\left[A_f^c\right]$ | Threshold of DnaA concentration | $10\,\mu m^{-3}$ | Fitted in this study |
| $\alpha_H$ | Extruder synthesis rate | $180\,h^{-1}\mu m^{-3}$ | Fitted in this study |
| $\eta$ | Relative noise level of $\lambda$ and $\alpha_A$ in stochastic model | 0.1 | *Fitted in this study* |
| **Switch model** | | | |
| $\left[D\right]_T$ | Concentration of total DnaA protein | $400\,\mu m^{-3}$ | |
| $\alpha_l$ | Activation rate of DnaA-ATP by lipid | $750\,h^{-1}$ | |
| $\alpha_{d1}$ | Activation rate of DnaA-ATP by the *DARS1* site | $100\,h^{-1}$ | |
| $\alpha_{d2}$ | Activation rate of DnaA-ATP by the *DARS2* site | $643\,h^{-1}$ | |
| $\beta_{datA}$ | Deactivation rate of DnaA-ATP by *data* site | $600\,h^{-1}$ | |
| $\beta_{rida}$ | Deactivation rate of DnaA-ATP by RIDA system | $500\,h^{-1}$ | |
| $K_D$ | Dissociation constant of DnaA activation and deactivation | $50\,\mu m^{-3}$ | |
| $K_D^P$ | Dissociation constant of DnaA promoter | $300\,\mu m^{-3}$ | |
| $f_C$ | Threshold of DnaA-ATP fraction for DNA replication initiation | 0.75 | *Berger and Wolde, 2022*; LDDR model |
| **Titration switch model and Titration-switch-extrusion model** | | | |

*Appendix 1—table 1 Continued on next page*

*Appendix 1—table 1 Continued*

| Parameter | Description | Values | Source |
|---|---|---|---|
| $K_D^P$ | Dissociation constant of DnaA promoter | $400\,\mu m^{-3}$ | *Berger and Wolde, 2022*; SI table switch-titration model |
| $n$ | Cooperativity of dnaA expression | 5 | |
| $\alpha_H$ | Extruder synthesis rate | $550\,h^{-1}\mu m^{-3}$ | Fitted in this study |
| $\left[A_{ATP}^{fc}\right]$ | Threshold of DnaA-ATP concentration for DNA replication initiation | $200\,\mu m^{-3}$ | Fitted in this study |

## Appendix 1—note 1

To understand the initiation mass dependency on *dnaA* expression level $\alpha_A$ (*Figure 2G*), we can solve the dynamics of DnaA amounts $A_{(t)}$ and cell volume $V_{(t)}$, yielding the solutions: $A_{(t)} = \frac{\alpha_A V_0}{\lambda} e^{\lambda t} + const$. The boundary conditions corresponding to successive rounds of DNA replication initiation at times $t = 0$ and $t = t_I$ are: $A_{(t=0)} = \frac{A_I \cdot O}{2} + \frac{A_b}{2}, V_{(t=0)} = \frac{V_I \cdot O}{2}$, and $A_{(t=t_I)} = A_I \cdot O + A_b, V_{(t=t_I)} = V_I \cdot O$, where $A_I$ is the threshold of DnaA number per *oriC* and $V_I$ is the cell volume per *oriC* at the time of initiation ($t_I$).

Applying these conditions, we derive

$$V_I = \frac{A_I - A_b}{\alpha_A}\lambda$$

showing that the initiation mass is inversely proportional to the DnaA expression level $\alpha_A$, assuming $A_I$ and $A_b$ are constant. This inverse relationship predicted by the titration model implies a much sharper dependence of initiation mass on DnaA expression than the exponential dependence observed experimentally.

Following the same analysis in the titration model, the extrusion model has the same DnaA amounts and volume dynamics as in the titration model with similar H-NS dynamics $H_{(t)} = \frac{\alpha_H V_0}{\lambda} e^{\lambda t} + const$. The condition of DNA replication initiation sets the boundary condition: $A_{(t=0)} = \frac{A_I \cdot O}{2} + \frac{A_b}{2} + H_{(t=0)}, V_{(t=0)} = \frac{V_I \cdot O}{2}$ and $A_{(t=t_I)} = A_I \cdot O + A_b + H_{(t=t_I)}, V_{(t=t_I)} = V_I \cdot O$.

Solving these equations, we get:

$$V_I = \frac{A_I - A_b}{\alpha_A + \alpha_H}\lambda$$

The addition of extruder shifts the inverse dependency of initiation mass-dnaA expression curve to the left, which results in a flatter curve. To achieve a better fit to the experiment data, $\alpha_H = 1500\,h^{-1}\mu m^{-3}$ were used in *Figure 6E*.

# Appendix 2

## Supplementary information for strains, plasmids, and oligonucleotides used in this study

**Appendix 2—table 1.** Strains used in this study.

| Strain | Relevant genetic marker(s) or features | Source or reference |
|---|---|---|
| MG1655 | *E. coli K12*(AMB1655) | *Liu et al., 2011* |
| CL1 | MG1655 Δ*cheZ*, Δ*lac* | *Liu et al., 2011* |
| MGCL1 | MG1655 Δ*lac* | This study |
| RdnaA1 | MG1655 $P_{dnaA}$-*dnaA*::$P_{kanR}$-*kanR yidA*::(*bla*:$P_{tet}$-*dnaA*)::*yidX intS*::$P_{tet}$-*tetR* | This study |
| RdnaA2 | RdnaA1 Δ*lac* | This study |
| RseqA1 | MG1655 *seqA*::*kan attB*::(*bla*:$P_{tet}$-*tetR-seqA*) | This study |
| Rhns1 | MG1655 *hns*::*kan attB*::(*bla*:$P_{tet}$-*tetR-hns*) | This study |
| Rhns2 | Rhns1 Δ*lac* | This study |
| CRidnaA1 | MG1655 *asnA*::($P_{J23119}$-*sgRNA*$_{dnaA}$:$P_{J23100}$-*tetR*:$P_{tet}$ -*dUn1Cas12f1*)::*viaA* | This study |

**Appendix 2—table 2.** Plasmids used in this study.

| Plasmid | Relevant genotype | Source or reference |
|---|---|---|
| pSIM5 | $Cm^r$, *repA101(Ts) ori*, λ Red | *Zheng et al., 2016* |
| plkml | $Amp^r$, pUC *ori*, *loxp-kan-loxp* | *Zheng et al., 2016* |
| pMD19-tetR | $Amp^r$, pUC *ori*, *bla*:$P_{tet}$-*tetR* | *Zheng et al., 2016* |
| pMD19-hupA-mcherry | $Amp^r$, pUC *ori*, *bla*:$P_{tet}$-*tetR-hupA-mcherry* | This study |
| pMD19-Rhns | $Amp^r$, pUC *ori*, *bla*:$P_{tet}$-*tetR-hns* | This study |
| pMD19-RseqA | $Amp^r$, pUC *ori*, *bla*:$P_{tet}$-*tetR-seqA* | This study |
| pMD19-RdnaA | $Amp^r$, pUC *ori*, *bla*:$P_{tet}$-*tetR-dnaA* | This study |
| p15A-RdnaA | $Kan^r$, p15A *ori*, $P_{tet}$-*tetR-dnaA* | This study |
| pTargetF | *aadA*$^r$, pMB1 *ori*, sgRNA | *Jiang et al., 2015* |
| pEcCas | $Kan^r$, pSC101 *ori*, *sacB* $P_{araB}$- λ Red $P_{cas}$-*cas9* | *Li et al., 2021* |
| pZA31-$P_{tet}$-M2-GFP | $Cm^r$, p15A ori, *Ptet-gfp* | *Liu et al., 2019* |
| CmPcas | $Cm^r$, pSC101 *ori*, *sacB* $P_{araB}$- λ Red $P_{cas}$-*cas9* | This study |
| CPP00458 | *aadA*$^r$, pSC101 *ori*, $P_{J23119}$-*sgRNA-T-*$P_{J23100}$-*tetR-T-*$P_{tet}$-*dUn1Cas12f1* | Gift from Xiongfei Fu lab |
| P_CRidnaA1 | *aadA*$^r$, pSC101 *ori*, $P_{J23119}$-*sgRNA*$_{dnaA}$-*T-*$P_{J23100}$-*tetR-T-*$P_{tet}$-*dUn1Cas12f1* | This study |
| pPT | $Cm^r$, pSC101 *ori*, Bsal-*lacZa*-Basl-*riboJ-sfgfp* | *Zong et al., 2017* |
| pPT-RFP | $Cm^r$, pSC101 *ori*, Bsal-*lacZa*-Basl-*riboJ-mcherry* | This study |
| pPT-lacZ | $Cm^r$, pSC101 *ori*, Bsal-*lacZa*-Basl-*riboJ-lacZ* | This study |
| $P_{syn66}$-GFP | $Cm^r$, pSC101 *ori*, $P_{syn66}$-*riboJ-sfgfp* | This study |
| $P_{con}$-GFP | $Cm^r$, pSC101 *ori*, $P_{con}$-*riboJ-sfgfp* | This study |
| $P_{syn66}$-RFP | $Cm^r$, pSC101 *ori*, $P_{syn66}$-*riboJ-mcherry* | This study |
| $P_{native}$-GFP | $Cm^r$, pSC101 *ori*, $P_{native}$-*riboJ-sfgfp* | This study |
| $P_{syn66}$-lacZ | $Cm^r$, pSC101 *ori*, $P_{syn66}$-*riboJ-lacZ* | This study |
| $P_{con}$-lacZ | $Cm^r$, pSC101 *ori*, $P_{con}$-*riboJ-lacZ* | This study |
| $P_{sny66}$-$P_{con}$-FPs | $Cm^r$, pSC101 *ori*, $P_{con}$-*riboJ-sfGFP*, $P_{syn66}$-*riboJ-mcherry* | This study |
| pET-28a-DnaA | $Kan^r$, pUC *ori f1 ori*, $P_{lacI}$-*lacI*, $P_{T7/lacO}$-*dnaA-6*his* | This study |
| pET-28a-H-NS | $Kan^r$, pUC *ori f1 ori*, $P_{lacI}$-*lacI*, $P_{T7/lacO}$-*hns-6*his* | This study |

**Appendix 2—table 3.** Primer pairs for synthetic reporter.

| Primers | Sequence | Use |
|---|---|---|
| DP420 | cctggtagatagattgacaagagttatccacagtaggatactgagcaca | |
| DP421 | agcttgtgctcagtatcctactgtggataactcttgtcaatctatctac | $P_{syn1}$ |
| DP422 | cctggtagatagattgacacttgttatacacagggcgatactgagcaca | |
| DP423 | agcttgtgctcagtatcgccctgtgtataacaagtgtcaatctatctac | $P_{syn2}$ |
| DP424 | cctggtagatagattgacaatactttttccacaggtagatactgagcaca | |
| DP425 | agcttgtgctcagtatctacctgtggaaaagtattgtcaatctatctac | $P_{syn3}$ |
| DP426 | cctggtagatagattgacaccgatcattcacagttagatactgagcaca | |
| DP427 | agcttgtgctcagtatctaactgtgaatgatcggtgtcaatctatctac | $P_{syn4}$ |
| DP428 | cctggtagatagattgacacttgtgtggataagggcgatactgagcaca | |
| DP429 | agcttgtgctcagtatcgcccttatccacacaagtgtcaatctatctac | $P_{syn5}$ |
| DP430 | cctggtagatagattgacattatccacatagttcccgatactgagcaca | |
| DP431 | agcttgtgctcagtatcgggaactatgtggataatgtcaatctatctac | $P_{syn6}$ |
| DP432 | cctggtagatagattgacacccctgcgattttttcccgatactgagcaca | |
| DP433 | agcttgtgctcagtatcgggaaaaatcgcaggggtgtcaatctatctac | $P_{syn7}$ |
| DP434 | cctgttatccacattgacaccccctgcgatagttcccgatactgagcaca | |
| DP435 | agcttgtgctcagtatcgggaactatcgcaggggtgtcaatgtggataa | $P_{syn8}$ |
| DP436 | cctggtagatagattgacacccctgcgatagttcccgatactttatccac | |
| DP437 | agctgtggataaagtatcgggaactatcgcaggggtgtcaatctatctac | $P_{syn9}$ |
| DP438 | cctgacagagttatccacagtagatagattgacaccgatcattcacagttagatactgagcaca | |
| DP439 | agcttgtgctcagtatctaactgtgaatgatcggtgtcaatctatctactgtggataactctgt | $P_{syn10}$ |
| DP440 | cctggaggggttatacacaactcaaagattgacaccgatcattcacagttagatactgagcaca | |
| DP441 | agcttgtgctcagtatctaactgtgaatgatcggtgtcaatctttgagttgtgtataacccctc | $P_{syn11}$ |
| DP442 | cctgccatactgtggaaaaggtagaagattgacaccgatcattcacagttagatactgagcaca | |
| DP443 | agcttgtgctcagtatctaactgtgaatgatcggtgtcaatcttctacctttttccacagtatgg | $P_{syn12}$ |
| DP444 | cctgttatccacattgacaccccctgcgatagttcccgatactttatccac | |
| DP445 | agctgtggataaagtatcgggaactatcgcaggggtgtcaatgtggataa | $P_{syn13}$ |
| DP446 | cctgttatccacattgacacccctgcgattttttcccgatactgagcaca | |
| DP447 | agcttgtgctcagtatcgggaaaaatcgcaggggtgtcaatgtggataa | $P_{syn14}$ |
| DP448 | cctgttatccacattgacattatccacatagttcccgatactgagcaca | |
| DP449 | agcttgtgctcagtatcgggaactatgtggataatgtcaatgtggataa | $P_{syn15}$ |
| DP450 | cctggtagatagattgacattatccacatagttcccgatactttatccac | |
| DP451 | agctgtggataaagtatcgggaactatgtggataatgtcaatctatctac | $P_{syn16}$ |
| DP452 | cctggtagatagattgacacccctgcgattttttcccgatactttatccac | |
| DP453 | agctgtggataaagtatcgggaaaaatcgcaggggtgtcaatctatctac | $P_{syn17}$ |
| DP454 | cctggatagattgacatgtggataagtgtggatgatactgagcaca | |
| DP455 | agcttgtgctcagtatcatccacacttatccacatgtcaatctatc | $P_{syn18}$ |
| DP456 | cctggatagattgacattatccacagttttcccgatactgagcaca | |
| DP457 | agcttgtgctcagtatcgggaaaactgtggataatgtcaatctatc | $P_{syn19}$ |

*Appendix 2—table 3 Continued on next page*

*Appendix 2—table 3 Continued*

| Primers | Sequence | Use |
|---|---|---|
| DP458 | cctggatagattgacattatccacagctttccagatactgagcaca | |
| DP459 | agcttgtgctcagtatctggaaagctgtggataatgtcaatctatc | $P_{syn20}$ |
| DP460 | cctgttatccacattgacacccctgcgatagttcccgatactgtggataa | |
| DP461 | agctttatccacagtatcgggaactatcgcaggggtgtcaatgtggataa | $P_{syn21}$ |
| DP462 | cctgtgtggataattgacacccctgcgatagttcccgatactttatccac | |
| DP463 | agctgtggataaagtatcgggaactatcgcaggggtgtcaattatccaca | $P_{syn22}$ |
| DP464 | cctgtgtggataattgacacccctgcgatagttcccgatactgtggataa | |
| DP465 | agctttatccacagtatcgggaactatcgcaggggtgtcaattatccaca | $P_{syn23}$ |
| DP466 | cctgtgtggataattgacacccctgcgattttttcccgatactgagcaca | |
| DP467 | agcttgtgctcagtatcgggaaaaatcgcaggggtgtcaattatccaca | $P_{syn24}$ |
| DP468 | cctgttatccacattgacatgtggataatagttcccgatactgagcaca | |
| DP469 | agcttgtgctcagtatcgggaactattatccacatgtcaatgtggataa | $P_{syn25}$ |
| DP470 | cctgtgtggataattgacattatccacatagttcccgatactgagcaca | |
| DP471 | agcttgtgctcagtatcgggaactatgtggataatgtcaattatccaca | $P_{syn26}$ |
| DP472 | cctgtgtggataattgacatgtggataatagttcccgatactgagcaca | |
| DP473 | agcttgtgctcagtatcgggaactattatccacatgtcaattatccaca | $P_{syn27}$ |
| DP474 | cctggtagatagattgacattatccacatagttcccgatactgtggataa | |
| DP475 | agctttatccacagtatcgggaactatgtggataatgtcaatctatctac | $P_{syn28}$ |
| DP476 | cctggtagatagattgacatgtggataatagttcccgatactttatccac | |
| DP477 | agctgtggataaagtatcgggaactattatccacatgtcaatctatctac | $P_{syn29}$ |
| DP478 | cctggtagatagattgacatgtggataatagttcccgatactgtggataa | |
| DP479 | agctttatccacagtatcgggaactattatccacatgtcaatctatctac | $P_{syn30}$ |
| DP480 | cctggtagatagattgacacccctgcgattttttcccgatactgtggataa | |
| DP481 | agctttatccacagtatcgggaaaaatcgcaggggtgtcaatctatctac | $P_{syn31}$ |
| DP482 | cctggtagatagattgacatgtggataatttttcccgatactgagcaca | |
| DP483 | agcttgtgctcagtatcgggaaaaattatccacatgtcaatctatctac | $P_{syn32}$ |
| DP484 | cctgtgtggataattgacattatccacatagttcccgatactttatccac | |
| DP485 | agctgtggataaagtatcgggaactatgtggataatgtcaattatccaca | $P_{syn33}$ |
| DP486 | cctgttatccacattgacattatccacatagttcccgatactgtggataa | |
| DP487 | agctttatccacagtatcgggaactatgtggataatgtcaatgtggataa | $P_{syn34}$ |
| DP488 | cctgtgtggataattgacattatccacatagttcccgatactgtggataa | |
| DP489 | agctttatccacagtatcgggaactatgtggataatgtcaattatccaca | $P_{syn35}$ |
| DP490 | cctgttatccacattgacatgtggataatagttcccgatactgtggataa | |
| DP491 | agctttatccacagtatcgggaactattatccacatgtcaatgtggataa | $P_{syn36}$ |
| DP492 | cctgtgtggataattgacatgtggataatagttcccgatactttatccac | |
| DP493 | agctgtggataaagtatcgggaactattatccacatgtcaattatccaca | $P_{syn37}$ |
| DP494 | cctgtgtggataattgacatgtggataatagttcccgatactgtggataa | |
| DP495 | agctttatccacagtatcgggaactattatccacatgtcaattatccaca | $P_{syn38}$ |

*Appendix 2—table 3 Continued on next page*

*Appendix 2—table 3 Continued*

| Primers | Sequence | Use |
|---------|----------|-----|
| DP496 | cctgttatccacattgacattatccacatagttcccgatactttatccac | |
| DP497 | agctgtggataaagtatcgggaactatgtggataatgtcaatgtggataa | $P_{syn39}$ |
| DP498 | cctgttatccacattgacacccctgcgattttttcccgatactttatccac | |
| DP499 | agctgtggataaagtatcgggaaaaatcgcaggggtgtcaatgtggataa | $P_{syn40}$ |
| DP500 | cctgttatccacattgacattatccacagttttcccgatactgagcaca | |
| DP501 | agcttgtgctcagtatcgggaaaactgtggataatgtcaatgtggataa | $P_{syn41}$ |
| DP502 | cctgttatccacattgacacccctttatccacacccgatactttatccac | |
| DP503 | agctgtggataaagtatcgggtgtggataaaggggtgtcaatgtggataa | $P_{syn42}$ |
| DP504 | cctgttatccacattgacacccctgcgttatccacagatactttatccac | |
| DP505 | agctgtggataaagtatctgtggataacgcaggggtgtcaatgtggataa | $P_{syn43}$ |
| DP506 | cctgttatccacattgacatgtggataatagttcccgatactttatccac | |
| DP507 | agctgtggataaagtatcgggaactattatccacatgtcaatgtggataa | $P_{syn44}$ |
| DP508 | cctgttatccacattgacacccctttgtggataacccgatactttatccac | |
| DP509 | agctgtggataaagtatcgggttatccacaaggggtgtcaatgtggataa | $P_{syn45}$ |
| DP510 | cctgttatccacattgacacccctgcgtgtggataagatactttatccac | |
| DP511 | agctgtggataaagtatcttatccacacgcaggggtgtcaatgtggataa | $P_{syn46}$ |
| DP512 | cctgttatccacattgacattttcccgatagttcccgatactttatccac | |
| DP513 | agctgtggataaagtatcgggaactatcgggaaaatgtcaatgtggataa | $P_{syn47}$ |
| DP514 | cctgttatccacattgacatcgggaaaatagttcccgatactttatccac | |
| DP515 | agctgtggataaagtatcgggaactattttcccgatgtcaatgtggataa | $P_{syn48}$ |
| DP516 | cctgttatccacattgacacccctttttcccgacccgatactttatccac | |
| DP517 | agctgtggataaagtatcgggtcgggaaaaaggggtgtcaatgtggataa | $P_{syn49}$ |
| DP518 | cctgttatccacattgacacccctcgggaaaacccgatactttatccac | |
| DP519 | agctgtggataaagtatcgggttttcccgaaggggtgtcaatgtggataa | $P_{syn50}$ |
| DP520 | cctgttatccacattgacacccctgcgtcgggaaaagatactttatccac | |
| DP521 | agctgtggataaagtatctttttcccgacgcaggggtgtcaatgtggataa | $P_{syn51}$ |
| DP522 | cctgtgtggataagatagattgacatgtggataagtgtggatgatactgagcaca | |
| DP523 | agcttgtgctcagtatcatccacacttatccacatgtcaatctatcttatccaca | $P_{syn52}$ |
| DP524 | cctgttatccacagatagattgacattatccacagctttccagatactgagcaca | |
| DP525 | agcttgtgctcagtatctggaaagctgtggataatgtcaatctatctgtggataa | $P_{syn53}$ |
| DP526 | cctgttatccacagatagattgacattatccacagttttcccgatactgagcaca | |
| DP527 | agcttgtgctcagtatcgggaaaactgtggataatgtcaatctatctgtggataa | $P_{syn54}$ |
| DP528 | cctgttatccacagatagattgacatgtggataagtgtggatgatactgagcaca | |
| DP529 | agcttgtgctcagtatcatccacacttatccacatgtcaatctatctgtggataa | $P_{syn55}$ |
| DP530 | cctgtgtggataattgacatgtggataagtgtggatgatactgagcaca | |
| DP531 | agcttgtgctcagtatcatccacacttatccacatgtcaattatccaca | $P_{syn56}$ |
| DP532 | cctgttgacattatccacagttttcccgatactttatccac | |
| DP533 | agctgtggataaagtatcgggaaaactgtggataatgtcaa | $P_{syn57}$ |

*Appendix 2—table 3 Continued on next page*

*Appendix 2—table 3 Continued*

| Primers | Sequence | Use |
|---------|----------|-----|
| DP534 | cctgttatccacattgacatgtggataattttttcccgatactttatccac | |
| DP535 | agctgtggataaagtatcgggaaaaattatccacatgtcaatgtggataa | $P_{syn58}$ |
| DP536 | cctgttatccacattgacattatccacattttttcccgatactgtggataa | |
| DP537 | agctttatccacagtatcgggaaaaatgtggataatgtcaatgtggataa | $P_{syn59}$ |
| DP538 | cctgtgtggataattgacattatccacattttttcccgatactttatccac | |
| DP539 | agctgtggataaagtatcgggaaaaatgtggataatgtcaattatccaca | $P_{syn60}$ |
| DP540 | cctgtgtggataattgacattatccacattttttcccgatactgtggataa | |
| DP541 | agctttatccacagtatcgggaaaaatgtggataatgtcaattatccaca | $P_{syn61}$ |
| DP542 | cctgtgtggataattgacatgtggataattttttcccgatactttatccac | |
| DP543 | agctgtggataaagtatcgggaaaaattatccacatgtcaattatccaca | $P_{syn62}$ |
| DP544 | cctgttatccacattgacatgtggataattttttcccgatactgtggataa | |
| DP545 | agctttatccacagtatcgggaaaaattatccacatgtcaatgtggataa | $P_{syn63}$ |
| DP546 | cctgtgtggataattgacatgtggataattttttcccgatactgtggataa | |
| DP547 | agctttatccacagtatcgggaaaaattatccacatgtcaattatccaca | $P_{syn64}$ |
| DP548 | cctgtgtggataattgacatgtggataagtgtggatgatacttgtggata | |
| DP549 | agcttatccacaagtatcatccacacttatccacatgtcaattatccaca | $P_{syn65}$ |
| DP550 | cctgttatccacattgacattatccacagtttttcccgatactttatccac | |
| DP551 | agctgtggataaagtatcgggaaaactgtggataatgtcaatgtggataa | $P_{syn66}$ |
| DP552 | cctgtgtggataattgacatgtggataagtgtggatgatactttatccac | |
| DP553 | agctgtggataaagtatcatccacacttatccacatgtcaattatccaca | $P_{syn67}$ |
| DP554 | cctgttgacacccctgcgatagttcccgatactgagcaca | |
| DP555 | agcttgtgctcagtatcgggaactatcgcaggggtgtcaa | $P_{con}$ |
| DP556 | cctgttatccacacccgggttatccacagtttttcccgagcccttatccac | |
| DP557 | agctgtggataagggctcgggaaaactgtggataacccgggtgtggataa | $P_{neg}$ |

**Appendix 2—table 4.** Oligonucleotides for the construction of strains and plasmids and qPCR.

| Primers | Sequence | Use |
|---------|----------|-----|
| DJP001 | gaaagaggagaaatactagatgaccatgattacggattcac | |
| DJP002 | ttgatgcctggcttatcattattttttgacaccagaccaact | Amplifying *lacZ* gene *from* MG1655 genome |
| DJP003 | gaaagaggagaaatactagatggtttccaagggcgagg | |
| DJP004 | ttgatgcctggcttatcattatttgtagagctcatccatgc | Amplifying mCherry gene *from* pMD19-*hupA-mcherry* |
| DJP005 | ccacaaggtctccagctgatcaagatcctgcaaaacgat | |
| DJP006 | ccacatggtctcccctgccaattttttgtctatggtcat | Amplifying native *dnaA* promoter (*P*$_{native}$) *from* MG1655 genome |
| DJP007 | ctgtttcttgcaagattactagtccatccagtgctcatttgtacagttcatccatacc | |
| DJP008 | ccttagtgactcctgcagtcctgggtgttgacaccctgcgat | Amplifying $P_{con}$-*gfp* cassette *from* $P_{con}$-GFP plasmid |

*Appendix 2—table 4 Continued on next page*

*Appendix 2—table 4 Continued*

| Primers | Sequence | Use |
|---|---|---|
| DJP009 | cgccatatgtcactttcgctttggca | Amplify *dnaA* gene *from* MG1655 genome for the construction of pET-28a-DnaA plasmid |
| DJP010 | cgcaagcttttacgatgacaatgttctga | |
| DJP011 | cgccatatgagcgaagcacttaaaat | Amplify *hns* gene *from* MG1655 genome for the construction of pET-28a-H-NS plasmid |
| DJP012 | cgcaagcttttattgcttgatcaggaaatc | |
| DJP013 | cgatctgcagaaagaggagaaatactaggtgtcactttcgctttggc | Amplifying *dnaA* gene *from* MG1655 genome |
| DJP014 | gtgagccggatccttacgatgacaatgttctgatt | |
| DJP015 | atgcctgcagtcacacaggaaacctactagatgaaacgattgaagttgatgatg | Amplifying *seqA* gene *from* MG1655 genome |
| DJP016 | gtcaggatccttagatagttccgcaaaccttct | |
| DJP017 | cagtaagcttaaagaggagaaatactagatgagcgaagcacttaaaattc | Amplifying *hns* gene *from* MG1655 genome |
| DJP018 | gtacggatccttattgcttgatcaggaaatcgtcg | |
| DJP019 | tggatcgcgaagaaaggc | Amplifying T3-$P_{tet}$-*tetR-dnaA* cassette *from* pMD19-RdnaA *plasmid* |
| DJP020 | tcgatatcaaccatggctgcggcaaaatcgctcgagt | |
| DJP021 | cgttttatttgatgggtcgacctgcagggaaagccacgt | Amplifying *kanr* gene from the plasmid pEcCas |
| DJP022 | ttcttcgcgatccatgctagcagcaaccaattaaccaattc | |
| DJP023 | gcagccatggttgatatcgagctcgcttgga | Amplifying p15A-$T_1$ fragment from the PZA31-$P_{tet}$-M2-GFP plasmid |
| DJP024 | cacatgaagtcgacccatcaaataaaacgaaaggctc | |
| DJP025 | tcaacccactgcagcaaccaattaaccaattctgattacgccccgccctgcca | *Amplifying Cm$^r$ gene from pZA31-$P_{tet}$-M2-GFP plasmid for the construction of* CmPcas |
| DJP026 | tgtctgcttacataaacagtaatacaaggggtgttatggagaaaaaaatcactgg | |
| DJP027 | ttaaaggtattaaaaacaactttttgtcttttaccttcccgtttcgctccaggaaacagctatgaccatg | Amplifying the $T_0$-Amp-$T_1$-$P_{tet}$-*tetR-hns* or $T_0$-Amp-$T_1$-$P_{tet}$-*tetR-seqA* cassette, and then assembling to *attB* locus via *λRed* recombination system |
| DJP028 | cacaggttgctccgggctatgaaatagaaaaatgaatccgttgaagcctgtgtaaaacgacggccagt | |
| DJP029 | tctattattacctcaacaaaccaccccaatataagtttgagattactacaatgattgaacaagatggattgcac | Amplifying *kan$^r$* gene from plkml plasmid for the deletion of native *hns* gene |
| DJP030 | aaaaaatcccgccgctggcgggatttttaagcaagtgcaatctacaaaagatcagaagaactcgtcaagaagg | |
| DJP031 | ggcctgcacgattgtggattgccattgctttgtccttgtctgcaacgttctagtgaacctcttcgaggg | Amplifying *kan$^r$* gene from plkml plasmid for the deletion of native *seqA* gene |
| DJP032 | catatactcctggcgacttgtattcagctaagacactgcactggattaaggccgatcatattcaataaccc | |
| DJP033 | gagtcatgcacagattcgta | *T0-$P_{tet}$-tetR-T3 cassette was amplified from plasmid pMD19-tetR with primers DJP035/DJP036, the intS upstream and downstream sequences were amplified from MG1655 genome with primers DJP033/DJP034, and DJP037/DJP038, respectively. Three fragments were ligated via overlap PCR with primers DJP033/DJP038.* |
| DJP034 | gatctgaagcgaaccatga | |
| DJP035 | tcatggttcgcttcagatcaggttgtgtgttcctcttcattc | |
| DJP036 | cctcatagccgatttgtttgaaggaaacagctatgaccatga | |
| DJP037 | caaacaaatcggctatgagg | |
| DJP038 | agtgtataagggtgttcagc | |

*Appendix 2—table 4 Continued on next page*

*Appendix 2—table 4 Continued*

| Primers | Sequence | Use |
|---|---|---|
| DJP039 | gtacgttagatcgtagacgcttggcgataaagaacgccacttcgcccggccgtgagcatttaggatccggctcaccttca | Amplifying T3-kanr-T1 cassette from p15A-RdnaA plasmid to replace the native $P_{native}$-*dnaA* on the genome |
| DJP040 | gatcgattaagccaattttttgtctatggtcattaaatttccaatatgcggcgtaaatctagggcggcggatttg | |
| DJP041 | ttaggcaccccaggctttac | The *dnaA*-$T_3$ and $T_0$-Amp-$T_1$-$P_{tet}$ fragments were amplified from the pMD19-RdnaA plasmid using primers DJP041/DJP042 and DJP043/DJP044, respectively. These two fragments were then combined via overlap PCR using primers DJP045/DJP046 to generate a homologous recombination fragment for inserting $T_0$-Amp -$T_1$-$P_{tet}$-*dnaA*-$T_3$ between the *yidA* and *yidX* genes. |
| DJP042 | cagtgatagagatactgagcacataagcttaaagaggagaaagactaggtgtcactttcgc | |
| DJP043 | gtgctcagtatctctatcactgatagggatgtcaatctctatcactgatagggagggactcgag | |
| DJP044 | aaaacgacggccagtgaa | |
| DJP045 | gatggcgtggcgtttgctattgagaagtatgtgctgaattaatctgtgggcggtcatcttcggctactgtct | |
| DJP046 | accgctgcaatttctggttgtatatgcagtaaaccaataatcagtaagcgcaggaaacagctatgaccatg | |
| DJP047 | saacttcgagtggagtccgccgtg | Inserting to the CPP00458 backbone to generate P_CRidnaA1 plasmid to shut down *dnaA* expression |
| DJP048 | cgagcacggcggactccactcgaa | |
| DJP049 | aaacgatgaagaccgtctttctcc | Amplifying homologous sequences to *asnA* gene |
| DJP050 | ttcttagacgtcaggtggcattattacagcagagaagggacg | |
| DJP051 | tgccacctgacgtctaagaa | Amplifying $P_{J23119}$-*sgRNA*$_{dnaA}$:$P_{J23100}$-*tetR*:$P_{tet}$-*dUn1Cas12f1* cassette from P_CRidnaA1 plasmid |
| DJP052 | tctagattactgcgcagatggcgacgataatgacagcagccaactcagcttc | |
| DJP053 | gccatctgcgcagtaatctagatcgcatcccggtatcaaagc | Amplifying homologous sequences to *viaA* gene |
| DJP054 | atgaacagtgtgcgaaagcg | |
| DJP055 | ccacaaggcatcgaacaagc | Assembling the upper three fragments through overlap PCR |
| DJP056 | gtggaacccggtactggaag | |
| DJP057 | cttctttggtgctgtactca | RT-qPCR primer for *rpoA* |
| DJP058 | tggttgatatcgagcaagtg | |
| DJP059 | cccgattgcaggatgagtt | RT-qPCR primer for *dnaA* |
| DJP060 | tacccaatcgaggacaaaac | |
| DJP061 | cgttatccggaccatatgaa | RT-qPCR primer for *gfp* |
| DJP062 | cttcaaatttcacttccgca | |
| DJP063 | aagttaaactgcgtggtact | RT-qPCR primer for *mcherry* |
| DJP064 | acaggtttcttggctttgta | |
| DJP065 | tatgttgaaattttccgccg | RT-qPCR primer for *seqA* |
| DJP066 | attcatccgaaagcagaagt | |
| DJP067 | gtattgacccgaacgaactg | RT-qPCR primer for *hns* |
| DJP068 | agtccaggtttttagtttcgc | |

**Appendix 2—table 5.** 43 3'-TAMRA probes spanning the whole coding sequence of the *dnaA* gene.

| | | | |
|---|---|---|---|
| gccaaagcgaaagtgacacg | agaacgataggtcggttctg | cttcctgagatcgttcttta | cacgaagtcgatggtgatcg |
| tgcaatcgggcaagacactg | acgtgtgtttgacgtttacg | ggcgttgaaggtgtggaaaa | tgaccagtttttcctgcaat |

*Appendix 2—table 5 Continued on next page*

*Appendix 2—table 5 Continued*

| | | | |
|---|---|---|---|
| aattctgtggctggtaactc | cgccagttggttagatttac | atctgttgattaccttccag | ccgtcttctgaatattgtcg |
| caatgggcgtatccacatac | ggaacaacgggttataggca | atagcgatccgaggtgagaa | cgcgactttgatcttgtagt |
| agcgtgttatcgctcagttc | atgcagcaggtgagttttac | caacgccgttgatctctttc | atcgacgcttggaaaggaga |
| cccaatcgaggacaaaacgg | taaaccactttggcattcgg | caaccgaagcgggatttcaa | tgtggttagtcagctctttc |
| ttattaaggtacttgtcccg | tgaacaaagcgctcggagtg | tttttcatcaggatcgccac | caccaaacgcatcgccaatc |
| gcagaaactggttagcagtc | tttgcagggctttaaccatg | acgaatgtcgttttcgtcgg | ttacggcaggcatgaagcac |
| cgacttcaaaacgcagctgt | ttaaactcttcgatcgcgtt | ggcgataaagaacgccactt | ctcttcacgcaactgctcga |
| gtagaaggcgcagcacgttg | atctacggaacggtagtagc | gtacgttagatcgtagacgc | aatcttctttgatatcgtgg |
| ggacgttatcccaacctgag | gaatatcgtcgatcagcagt | ggtaaagttggcattggcaa | |

