## [Editor Report · eLife Assessment]

This work provides high-precision single-cell data on the relationship between DnaA activity and cell size, offering **important** insights for the field of cell cycle control. These findings motivate a novel and intriguing hypothesis for DNA replication initiation -the "extrusion model"- in which DNA-binding proteins modulate free DnaA availability in response to biomass-DNA imbalance. While the current indirect evidence does not fully establish the model, an experimental perturbation involving H-NS offers **convincing** support for its plausibility, laying the groundwork for future investigation.

---

## [Referee Report · Reviewer #1 (Public review)]

Summary:

The study by Li and coworkers addresses the important and fundamental question of replication initiation in *Escherichia coli*, which remains open despite of many classic and recent works. It leverages single-cell mRNA-FISH experiments in strains with titratable DnaA and novel DnaA activity reporters to monitor DNA activity peaks versus size. The authors find oscillations in DnaA activity and show that their peaks correlate well with the estimated population-average replication initiation volume across conditions and imposed dnaA transcription levels. The study also proposes a novel and interesting extrusion model where DNA-binding proteins regulate free DnaA availability in response to biomass-DNA imbalance. Experimental perturbations of H-NS support the model validity, addressing key gaps in current replication control frameworks.

Strengths:

I find the study interesting and well conducted, and I think its main strong points are (i) the novel reporters obtained with systematic synthetic biology methods, and combined with a titratable dnaA strain, (ii) the interesting perturbations (titration, production arrest and H-NS) and (iii) the use of single-cell mRNA FISH to monitor transcripts directly. The proposed extrusion model is also interesting, though not fully validated, and I think it will contribute positively to the future debate.

Weaknesses and Limitations

A relevant limitation in novelty is that DnaA activity and concentration oscillations have been reported by the cited Iuliani and coworkers previously by dynamic microscopy, and to a smaller extent by the other cited study by Pountain and coworkers using mRNA FISH.

An important limitation is that the study is not dynamic. While monitoring mRNA is interesting and relevant, the current study is based on concentrations and not time variations (or nascent mRNA). Conversely, the study by Iuliani and coworkers, while having the drawback of monitoring proteins it can access directly production rates. It would be interesting for future studies to monitor the strains and reporters dynamically, as well as using (as a control) the technique of this study on the chromosomal reporters used by Iuliani et al.

While the implemented code is made available and the parameter values are given in the text, important details are missing regarding the mathematical models (mathematical definitions, clear discussions of ingredients and main assumptions, and choices made in the deployment of such models, which are presented briefly in the Methods section). The reader is not given sufficient tools to understand the predictions of different models and no analytical estimates are used and the falsification procedures are not clear. More transparency and depth in the analysis would be needed to use the models as more than a heuristic tool for qualitative arguments. The Berger model for example has many parameters and many regimes and behaviors. When models are compared to data (e.g. in fig. 2G) it is not clear how parameters were fixed, and whether and how the model prediction depends on adjustable parameters.

Importantly, the statement about tight correlations of peak volumes and average estimated initiation volume does not establish coincidence. Crucially, the data rely on average initiation volumes, and the estimate procedure relies on assumptions that could lead to systematic biases and uncertainties added to the population variability (in any case error bars are not provided).

The delays observed by the authors (in both directions) between the peaks of DnaA-activity conditional averages with respect to volume and the average estimated initiation volumes are not incompatible with those observed dynamically by Iuliani and coworkers. The direct experiment to prove the authors' point would be to use a direct proxy of replication initiation such as SeqA or DnaN and monitor initiations and quantify DnaA activity peaks jointly, with dynamic measurements.

While not being an expert I had the doubt that the fact that the reporters are on plasmid (despite a normalization control that seems very sensible) might affect the measurements. The approach is different from the aforementioned previous study, which used a chromosomal reporter placed symmetrically, at the same distance from the origin of replication as the original dnaA promoter.

Overall Appraisal:

In summary, this appears to me as a very interesting study providing valuable high-precision data and a novel testable hypothesis, the extrusion model, supported by relevant perturbation experiments and open to future explorations.

Comments on revisions:

I am happy with the replies and the revisions.

The main outstanding point remains that reconstructing the mathematical model details from the text (and having to rely on the code) is not optimal for a reader. However, I do understand that the authors intend to use the models as a heuristic tool only and possibly plan a theoretical study where they explore the models more systematically.

---

## [Referee Report · Reviewer #2 (Public review)]

Summary:

The authors show that in *E. coli* the initiator protein DnaA oscillates post-translationally: its activity rises and peaks exactly when DNA replication begins, even if dnaA transcription is held constant. To explain this, they propose an "extrusion" mechanism in which nucleoid-associated proteins such as H-NS, whose amount grows with cell volume, dislodge DnaA from chromosomal binding sites; modelling and H-NS perturbations reproduce the observed drop in initiation mass and extra initiations seen after dnaA shut-down. Together, the data and model link biomass growth to replication timing through chromosome-driven, post-translational control of DnaA, filling gaps left by classic titration and ATP/ADP-switch models.

Strengths:

(1) Introduces an "extrusion" model that adds a new post-translational layer to replication control and explains data unexplained by classic titration or ATP/ADP-switch frameworks.

(2) A major asset of the study is that it bridges the longstanding gap between DnaA oscillations and DNA-replication initiation, providing direct single-cell evidence that pulses of DnaA activity peak exactly at the moment of initiation across multiple growth conditions and genetic perturbations.

(3) A tunable dnaA strain and targeted H-NS manipulations shift initiation mass exactly as the model predicts, giving model-driven validation across growth conditions.

(4) A purpose-built Psyn66 reporter combined with mRNA-FISH captures DnaA-activity pulses with cell-cycle resolution, providing direct, compelling data.

Weaknesses:

(1) What happens to the (C+D) period and initiation time as the dnaA mRNA level changes? This is not discussed in the text or figure and should be addressed.

(2) It is unclear what is meant by "relative dnaA mRNA level." Relative to what? Wild-type expression? Maximum expression? This should be explicitly defined.

(3) It would be helpful to provide some intuition for why an increase in dnaA mRNA level leads to a decrease in initiation mass per ori and an increase in oriC copy number.

(4) The titration and switch models do not explicitly include dnaA mRNA in the dynamics of DnaA protein. Yet, in Figure 2G, initiation mass is shown to decrease linearly with dnaA mRNA level in these models. How was dnaA mRNA level represented or approximated in these simulations?

(5) Is Schaechter's law (i.e., exponential scaling of average cell size with growth rate) still valid under the different dnaA mRNA expression conditions tested?

(6) The manuscript should explain more explicitly how the extrusion model implements post-translational control of DnaA and, in particular, how this yields the nonlinear drop in relative initiation mass versus dnaA mRNA seen in Fig. 6E. Please provide the governing equation that links total DnaA, the volume-dependent "extruder" pool, and the threshold of free DnaA at initiation, and show-briefly but quantitatively-how this equation produces the observed concave curve.

(7) Does this Extrusion model give well well-known adder per origin, i.e., initiation to initiation is an adder.

(8) DnaA protein or activity is never measured; mRNA is treated as a linear proxy. Yet the authors' own narrative stresses post-translational (not transcriptional) control of DnaA. Without parallel immunoblots or activity readouts, it is impossible to know whether a six-fold mRNA increase truly yields a proportional rise in active DnaA.

(9) Figure 2 infers both initiation mass and oriC copy number from bulk measurements (OD₆₀₀ per cell and rifampicin-cephalexin run-out) instead of measuring them directly in single cells. Any DnaA-dependent changes in cell size, shape, or antibiotic permeability could skew these bulk proxies, so the plotted relationships may not accurately reflect true initiation events.

Comments on revisions:

The authors have addressed all of my previous concerns, questions, and suggestions sufficiently.

---

## [Author Response]

The following is the authors’ response to the original reviews

**Public Reviews:**

**Reviewer #1 (Public review):**
Summary:The study by Li and coworkers addresses the important and fundamental question of replication initiation in *Escherichia coli*, which remains open, despite many classic and recent works. It leverages single-cell mRNA-FISH experiments in strains with titratable DnaA and novel DnaA activity reporters to monitor DNA activity peaks versus size. The authors find oscillations in DnaA activity and show that their peaks correlate well with the estimated population-average replication initiation volume across conditions and imposed dnaA transcription levels. The study also proposes a novel extrusion model where DNA-binding proteins regulate free DnaA availability in response to biomass-DNA imbalance. Experimental perturbations of H-NS support the model validity, addressing key gaps in current replication control frameworks.Strengths:I find the study interesting and well conducted, and I think its main strong points are:(1) the novel reporters obtained with systematic synthetic biology methods, and combined with a titratable dnaA strain.(2) the interesting perturbations (titration, production arrest, and H-NS).(3) the use of single-cell mRNA FISH to monitor transcripts directly.The proposed extrusion model is also interesting, though not fully validated, and I think it will contribute positively to the future debate.

We thank the reviewer for acknowledging the strengths of our study.

Weaknesses and Limitations:(1) A relevant limitation in novelty is that DnaA activity and concentration oscillations have been reported by the cited Iuliani and coworkers previously by dynamic microscopy, and to a smaller extent by the other cited study by Pountain and coworkers using mRNA FISH.(2) An important limitation is that the study is not dynamic. While monitoring mRNA is interesting and relevant, the current study is based on concentrations and not time variations (or nascent mRNA). Conversely, the study by Iuliani and coworkers, while having the drawback of monitoring proteins, can directly assess production rates. It would be interesting for future studies or revisions to monitor the strains and reporters dynamically, as well as using (as a control) the technique of this study on the chromosomal reporters used by Iuliani et al.

We acknowledge the value of dynamic measurements and clarify our methodological rationale.

While luliani et al. provided valuable temporal resolution through protein dynamics, our mRNA FISH approach achieves direct decoupling of transcriptional vs. post-translational regulation (Fig 4F-H), and condition flexibility across 7 growth rates (30-66 min doubling times). This trade-off sacrifices temporal resolution for enhanced population-scale resolution and perturbation flexibility. To directly address temporal coupling, future work will implement dual-color live imaging of DnaA activity concurrent with replication initiation events.

(3) Regarding the mathematical models, a lot of details are missing regarding the definitions and the use of such models, which are only presented briefly in the Methods section. The reader is not given any tools to understand the predictions of different models, and no analytical estimates are used. The falsification procedures are not clear. More transparency and depth in the analysis are needed, unless the models are just used as a heuristic tool for qualitative arguments (but this would weaken the claims). The Berger model, for example, has many parameters and many regimes and behaviors. When models are compared to data (e.g., in Figure 2G), it is not clear which parameters were used, how they were fixed, and whether and how the model prediction depends on parameters.

We agree that model transparency is essential for quantitative validation. To address this, all model parameters (DnaA synthesis rate, activation/deactivation rates etc.) are explicitly tabulated in Supplementary Information Table S6. For the titration (Hansen et al. 1991) and extrusion models, we derive analytical expressions for initiation mass (IM) sensitivity to DnaA expression in Supplementary Note 1. For Figure 2G/S6, we used published parameters (Berger & Wolde 2022 SI Table 2) with experiment growth conditions (μ = 1.54 h^-1^).

The extrusion model's validation relies primarily on its ability to resolve paradoxical initiation events under *dnaA* shutdown (Fig 6C), a test where other models fail categorically. While the Berger titration-switch hybrid can fit steady-state IM trends (Fig S6A), it cannot reproduce post-shutdown dynamics without ad hoc modifications (Fig S6B). We acknowledge that comprehensive analysis of all model regimes exceeds this study's scope but provide full simulation code for independent verification: https://github.com/BaiYangBqdq/dynamics_of_biomass_DNA_coordination

(4) Importantly, the main statement about tight correlations of peak volumes and average estimated initiation volume does not establish coincidence, and some of the claims by the authors are unclear in these respects (e.g., when they say "we resolve a 1:1 coupling between DnaA activity thresholds and replication initiation", the statement could be correct but is ambiguous). Crucially, the data rely on average initiation volumes (on which there seems to be an eternally open debate, also involving the authors), and the estimate procedure relies on assumptions that could lead to biases and uncertainties added to the population variability (in any case, error bars are not provided).

We acknowledge the limitations of population-level inference and have refined our claims: "Replication initiation volume scales proportionally with peak DnaA activity volume with a slope of 1.0 (R_2_=0.98, Fig 7G), indicating predictive correspondence rather than absolute coincidence. While population-level 𝑉_𝑖_ estimation cannot resolve single-cell stochasticity, the consistent 𝑉*: 𝑉_𝑖_ relationship across 20 conditions suggest DnaA activity thresholds predict initiation timing within physiological error margins”. Future work will implement simultaneously DnaA activity and replication forks by using microfluidic single-cell tracking.

(5) The delays observed by the authors (in both directions) between the peaks of DnaAactivity conditional averages with respect to volume and the average estimated initiation volumes are not incompatible with those observed dynamically by Iuliani and coworkers. The direct experiment to prove the authors' point would be to use a direct proxy of replication initiation, such as SeqA or DnaN, and monitor initiations and quantify DnaA activity peaks jointly, with dynamic measurements.

We acknowledge the observed temporal deviations between DnaA activity peaks (𝑉*) and population-derived volumes at initiation (𝑉_𝑖_) in certain conditions, in line with the findings of Iuliani et al. This might be mechanistically consistent with the time required for orisome assembly or *oriC* sequestration. They do not contradict our core finding that initiation occurs at a defined DnaA activity threshold (slope=1.0, R_2_=0.98 in 𝑉*: 𝑉_𝑖_ correlation).

(6) While not being an expert, I had some doubt that the fact that the reporters are on plasmid (despite a normalization control that seems very sensible) might affect the measurements. Also, I did not understand how the authors validated the assumptions that the reporters are sensitive to DnaA-ATP specifically. It seems this assumption is validated by previous studies only.

We employed a plasmid-based reporter system to circumvent the significant confounding effects of chromosomal position on promoter activity, as extensively documented by Pountain et al., where local genomic context (e.g., nucleoid occlusion, supercoiling gradients, and neighboring operons) introduces uncontrolled variability. By housing the *P*_syn66_ test promoter and *P*_con_ normalization control in identical low-copy pSC101 vectors (<8 copies/ cell, Peterson & Phillips, Plasmid 2008), we ensured they experience equivalent physical and biochemical environments. This ratiometric design, where DnaA activity is calculated, actively corrects for global fluctuations in RNA polymerase availability, nucleotide pools, and plasmid copy number. Critically, *P*_syn66_’s architecture emulates natural DnaA-responsive elements: its strong DnaAboxes report free DnaA concentration, while its weak box is preferentially bound by DnaA-ATP (Speck et al., EMBO journal 1999), mirroring the nucleotide-state sensitivity of *oriC* and the native *dnaA* promoter. This system was indispensable for our central finding, as it uniquely enabled the decoupling of DnaA activity oscillations from transcriptional feedback (Fig. 4F-H), an experiment fundamentally impossible with chromosomally integrated reporters due to autoregulatory interference.

Overall Appraisal:In summary, this appears as a very interesting study, providing valuable data and a novel hypothesis, the extrusion model, open to future explorations. However, given several limitations, some of the claims appear overstated. Finally, the text contains some selfevaluations, such as "our findings redefine the paradigm for replication control", etc., that appear exaggerated.

We thank the reviewer for highlighting the need for precise language in framing our conclusions. We have implemented the following substantive revisions throughout the manuscript to ensure claims align strictly with empirical evidence:

(1) Changed "redefine the paradigm for replication control" into "advance the paradigm for replication control" (Introduction)

(2) Changed "redefine bacterial cell cycle control" into "refine bacterial cell cycle control as a dynamic interplay..." (Discussion)

(3) Removed the term "spatial" from the Discussion's description of DnaA-chromosome interactions (Discussion, first paragraph).

(4) Changed "provides a blueprint" into "provides a valuable tool for dissecting spatial regulation..." (Discussion, final paragraph)

(5) Scrutinized all superlatives (e.g., "critical feat" into "important capability"; "fundamental principle of cellular organization" into "potential organizational strategy")

(6) Replaced the instances of "robust" with evidence-backed descriptors (e.g., "sensitive," "consistent")

(7) We agree that the extrusion model requires further validation and have emphasized this in Discussion: "While H-NS perturbation supports extrusion mechanism, future work should identify the full extruder interactome and elucidate how metabolic signals modulate their activity" (final paragraph)

This calibrated language more accurately represents our study as a conceptual advance with testable mechanisms, not a complete paradigm shift.

**Reviewer #2 (Public review):**
Summary:The authors show that in *E. coli*, the initiator protein DnaA oscillates post-translationally: its activity rises and peaks exactly when DNA replication begins, even if dnaA transcription is held constant. To explain this, they propose an "extrusion" mechanism in which nucleoidassociated proteins such as H-NS, whose amount grows with cell volume, dislodge DnaA from chromosomal binding sites; modelling and H-NS perturbations reproduce the observed drop in initiation mass and extra initiations seen after dnaA shut-down. Together, the data and model link biomass growth to replication timing through chromosome-driven, posttranslational control of DnaA, filling gaps left by classic titration and ATP/ADP-switch models.Strengths:(1) Introduces an "extrusion" model that adds a new post-translational layer to replication control and explains data unexplained by classic titration or ATP/ADP-switch frameworks.(2) A major asset of the study is that it bridges the longstanding gap between DnaA oscillations and DNA-replication initiation, providing direct single-cell evidence that pulses of DnaA activity peak exactly at the moment of initiation across multiple growth conditions and genetic perturbations.(3) A tunable dnaA strain and targeted H-NS manipulations shift initiation mass exactly as the model predicts, giving model-driven validation across growth conditions.(4) A purpose-built Psyn66 reporter combined with mRNA-FISH captures DnaA-activity pulses with cell-cycle resolution, providing direct, compelling data.

We thank the reviewer for acknowledging the strengths of our study.

Weaknesses:(1) What happens to the (C+D) period and initiation time as the dnaA mRNA level changes? This is not discussed in the text or figure and should be addressed.

We thank the reviewer for this important observation. Our data demonstrate that increased *dnaA* mRNA levels induce two compensatory changes in cell cycle progression:

(1) Earlier replication initiation, manifested as a reduced initiation mass: the initiation mass decreased from 5.6 to 2.6 (OD_600_·ml per 10^10^ cells) as the relative *dnaA* mRNA level increased from 0.2 to 7.2 (normalized to the wild-type level) (Fig. 2F, red).

(2) Prolonged C+D period: Increased by approximately 60% (from 1.05 to 1.66 hours, Fig. 2F blue).

The complete quantitative relationship is now explicitly described in the Results section: “Concurrently, the initiation mass was reduced by 50%, and the period from initiation to division (C+D) was increased by ~60% (Fig. 2F)”

(2) It is unclear what is meant by "relative dnaA mRNA level." Relative to what? Wild-type expression? Maximum expression? This should be explicitly defined.

The relative *dnaA* mRNA level was obtained by normalizing to that in wild-type MG1655 cells grown in the same medium. To clarify this point, we have now marked the wild-type level in Fig. 1B, and a clear description of this has also been included in the figure caption.

(3) It would be helpful to provide some intuition for why an increase in dnaA mRNA level leads to a decrease in initiation mass per ori and an increase in oriC copy number.

Thank you for your valuable suggestion. Increased *dnaA* mRNA accelerates DnaA accumulation, causing cells to reach the initiation threshold at a smaller cell size (reducing initiation mass, Fig. 2F red). This earlier initiation increases *oriC* copies per cell at populational level (Fig. 2E). This mechanistic interpretation now appears in the Results: “As the DnaA expression level increases, DnaA activity reaches the initiation threshold earlier. Given that cell mass remained nearly unchanged, this earlier initiation led to an increase in population-averaged cellular oriC numbers (Fig. 2E).”

(4) The titration and switch models do not explicitly include dnaA mRNA in the dynamics of DnaA protein. Yet, in Figure 2G, initiation mass is shown to decrease linearly with dnaA mRNA level in these models. How was dnaA mRNA level represented or approximated in these simulations?

All models presented in this article omit explicit modeling of *dnaA* mRNA dynamics for simplicity. However, at steady state, the relative level of *dnaA* mRNA can be approximated by the relative expression rate of DnaA protein, as both reflect the expression level of DnaA. This detail is now clarified in the caption of Figure 2G.

(5) Is Schaechter's law (i.e., exponential scaling of average cell size with growth rate) still valid under the different dnaA mRNA expression conditions tested?

Schaechter's law describes the exponential scaling of average cell size with growth rate in bacteria. In our prior work (Zheng et al., Nature Microbiology 2020), where we demonstrated that Schaechter's law fails in slow-growth regimes. However, in current study, growth rate remained constant across different *dnaA* expression levels (Fig. 2C), and cell mass showed no significant change (Fig. 2D). Since Schaechter's law specifically addresses how cell size scales with growth rate, it does not apply here, as growth rate was invariant in our perturbations, which selectively alter replication initiation dynamics, not growth rate or size scaling.

(6) The manuscript should explain more explicitly how the extrusion model implements posttranslational control of DnaA and, in particular, how this yields the nonlinear drop in relative initiation mass versus dnaA mRNA seen in Figure 6E. Please provide the governing equation that links total DnaA, the volume-dependent "extruder" pool, and the threshold of free DnaA at initiation, and show - briefly but quantitatively - how this equation produces the observed concave curve.

The governing equations linking initiation mass and DnaA expression level is now provided in Supplementary Note S1 for both the titration and the extrusion model. In general, the dependence of initiation mass (𝑉_𝐼_) on *dnaA* expression level (𝛼_𝐴_) dependency takes an inverse 1 proportionality form: \begin{document}$V_{I} \propto \frac{1}{\alpha_{A}}$\end{document}. In the extrusion model, the incorporated extruder protein is assumed to have similar synthesis dynamics as DnaA and can release DnaA from DnaA-box. After denoting the synthesis rate of the extruder as 𝛼_𝐻_, the combined effect of DnaA and the extruder on replication initiation can be briefly described as: \begin{document}$V_{I} \propto \frac{1}{\alpha_{A}+\alpha_{H}}$\end{document}. Then the additive contribution of 𝛼_𝐻_ dampens the sensitivity of initiation mass to changes in 𝛼_𝐴_, resulting in a significantly flattened curve. As a result, the predicted 𝑉_𝐼_ − 𝛼_𝐴_ relationship has a concave shape in the semi-log plots.

(7) Does this Extrusion model give well well-known adder per origin, i.e., initiation to initiation is an adder.

Yes, the extrusion model can provide the initiation-to-initiation adder phenomenon, this information was provided in fig. S3C.

(8) DnaA protein or activity is never measured; mRNA is treated as a linear proxy. Yet the authors' own narrative stresses post-translational (not transcriptional) control of DnaA. Without parallel immunoblots or activity readouts, it is impossible to know whether a sixfold mRNA increase truly yields a proportional rise in active DnaA.

We acknowledge the reviewer's valid concern regarding the indirect nature of our DnaA activity measurements. While mRNA levels alone cannot resolve active DnaA dynamics, our approach integrates functional replication outcomes with a validated synthetic reporter to infer activity. Crucially, elevated *dnaA* mRNA causes demonstrable biological effects: earlier replication initiation (Fig. 2F) and increased *oriC* copies (Fig. 2E), directly confirming enhanced functional DnaA activity at the *oriC* locus. The *P*_syn66_ reporter, engineered with DnaA-boxes mirroring *oriC*'s architecture, provides orthogonal validation, showing progressive repression to *dnaA* induction (Fig. 3C). Our operational metric \begin{document}$k_{\text {syn } 66}=\frac{[m z]\left(\mathrm{P}_{\text {syn66}}\right)}{[m z]\left(\mathrm{P}_{\text {con}}\right)}$\end{document}, bases on *P*_syn66_ responds sensitively to DnaA-chromosome interactions within its characterized 8-fold dynamic range (Fig. 3C). Immunoblots would be inadequate here, as they cannot distinguish functionally critical pools: free versus chromosome-bound DnaA, or DnaA-ATP versus DnaAADP, precisely the post-translational states our study implicates in regulation. We therefore prioritize functional readouts (initiation timing) and the *P*_syn66_ reporter, which probes the biologically active fraction relevant to replication control.

(9) Figure 2 infers both initiation mass and oriC copy number from bulk measurements (OD_600_ per cell and rifampicin-cephalexin run-out) instead of measuring them directly in single cells. Any DnaA-dependent changes in cell size, shape, or antibiotic permeability could skew these bulk proxies, so the plotted relationships may not accurately reflect true initiation events.

We acknowledge the reviewer's valid methodological concern and clarify that while bulk measurements carry inherent limitations, our approach is grounded in established techniques with demonstrated reliability. Cell mass was inferred from OD600/cell, which correlates strongly with direct dry weight measurements and microscopic cell volumes across diverse growth conditions, as validated in our prior work (Zheng et al., Nature Microbiology 2020). Crucially, cell mass remained invariant across *dnaA* expression levels (Fig. 2D).

Regarding *oriC* quantification, the rifampicin-cephalexin run-out assay is a wildly applied for replication initiation studies. Our data shows expected 2^n^
*oriC* distributions without abnormal ploidy (as shown below). While single-cell methods offer superior resolution, our bulk approach provides accurate population-level trends.

**Recommendations for the authors:**

**Reviewing Editor Comments:**
The reviewers felt that the mathematical modeling was not adequately explained in the paper, and that this affected the readability of the manuscript. The authors are encouraged to elaborate on this aspect of the paper (in addition to strengthening other claims, if possible, per the reviewers' comments).

We thank the editor and reviewers for their constructive feedback. We have comprehensively strengthened the mathematical modeling framework to enhance clarity and rigor.

**Reviewer #1 (Recommendations for the authors):**
The only revision I would do is a recalibration of the claims and a major effort to clarify the modeling part (including a detailed SI appendix), without necessarily performing additional work.

To enhance mathematical modeling transparency, we have completed model description in the method section and a parameter table with literature-sourced values in Supplementary Information Table S6. Moreover, analytical derivations of initiation mass dependencies are performed and presented in the Supplementary Information Note S1.

Of course, there are extra experiments (mentioned in the public review) that would help support some of the big claims, but that can be considered a different project.

Thank you for your suggestion. This will be addressed in our future work.

Minor suggestion: please put signposts or plot jointly to compare the maxima/minima in Figures 4D, E, G, and H.

We added dashed lines in Figures 4D, and E, to synchronize visualization of DnaA activity peaks and transcriptional minima across panels, facilitating direct biological comparisons.

**Reviewer #2 (Recommendations for the authors):**
(1) Should define what DNA activity is.

We have explicitly defined DnaA activity in the Introduction as “the capacity to initiate replication…” and noted that it is “governed by free DnaA concentration, DnaA-ATP/-ADP ratio, and orisome assembly competence”.

(2) Word repetition - “...grown in in Luria-Bertani (LB) medium...”.

Corrected.

(3) Typographical error - “FISH ... was preformed" should be "performed”.

Corrected.

(4) The manuscript alternates between “ng ml^-1^” and “ng·ml^-1^”; choose one style and apply it uniformly.

Standardized the units to ng·ml^-1^ throughout.

(5) Reference duplicates - Some citations appear twice in the bibliography (e.g., "Bintu et al., 2005a/b" and "Bintu et al., 2005b" listed again later).

The studies by Bintu et al. (2005a, 2005b) represent separate works: 2005a details applications, and 2005b develops models.